# Maternal Phylogenetic Relationships and Genetic Variation among Rare, Phenotypically Similar Donkey Breeds

**DOI:** 10.3390/genes12081109

**Published:** 2021-07-22

**Authors:** Andrea Mazzatenta, Massimo Vignoli, Maurizio Caputo, Giorgio Vignola, Roberto Tamburro, Francesco De Sanctis, Jordi Mirò Roig, Roberta Bucci, Domenico Robbe, Augusto Carluccio

**Affiliations:** 1Faculty of Veterinary Medicine, University of Teramo, SP 18, 64100 Teramo, Italy; mvignoli@unite.it (M.V.); mcaputo@unite.it (M.C.); rtamburro@unite.it (R.T.); rbucci@unite.it (R.B.); drobbe@unite.it (D.R.); acarluccio@unite.it (A.C.); 2Physiology and Physiopathology Section, Neuroscience, Imaging and Clinical Sciences Department, University G. D’Annunzio of Chieti-Pescara, Via Dei Vestini, 31, 66100 Chieti, Italy; 3Section of Immunology, Department of Medicine, University of Verona, Piazzale L.A. Scuro 10, 37134 Verona, Italy; Francesco.desanctis@univr.it; 4Servei de Reproducció Equina, Departament de Medicina i Cirurgia Animals, Autonomous University of Barcelona, Edifici V, 08193 Bellaterra, Spain; jordi.miro@uab.es

**Keywords:** genetics, mitochondrial DNA, donkey, Martina Franca, Ragusano, Pantesco, Catalonian

## Abstract

The mitochondrial DNA (mtDNA) D-loop of endangered and critically endangered breeds has been studied to identify maternal lineages, characterize genetic inheritance, reconstruct phylogenetic relations among breeds, and develop biodiversity conservation and breeding programs. The aim of the study was to determine the variability remaining and the phylogenetic relationship of Martina Franca (MF, with total population of 160 females and 36 males), Ragusano (RG, 344 females and 30 males), Pantesco (PT, 47 females and 15 males), and Catalonian (CT) donkeys by collecting genetic data from maternal lineages. Genetic material was collected from saliva, and a 350 bp fragment of D-loop mtDNA was amplified and sequenced. Sequences were aligned and evaluated using standard bioinformatics software. A total of 56 haplotypes including 33 polymorphic sites were found in 77 samples (27 MF, 22 RG, 8 PT, 19 CT, 1 crossbred). The breed nucleotide diversity value (π) for all the breeds was 0.128 (MF: 0.162, RG: 0.132, PT: 0.025, CT: 0.038). Principal components analysis grouped most of the haplogroups into two different clusters, I (including all haplotypes from PT and CT, together with haplotypes from MF and RG) and II (including haplotypes from MF and RG only). In conclusion, we found that the primeval haplotypes, haplogroup variability, and a large number of maternal lineages were preserved in MF and RG; thus, these breeds play putative pivotal roles in the phyletic relationships of donkey breeds. Maternal inheritance is indispensable genetic information required to evaluate inheritance, variability, and breeding programs.

## 1. Introduction

The genus *Equus* is the only remaining member of the family *Equidae*, which includes both extant and fossil species [1]. The noncaballine forms include the African wild ass *Equus africanus*; zebras *Equus quagga* (formerly *Equus burchellii*), *Equus grevyi*, and *Equus zebra* (with two subspecies *Equus zebra zebra* of South Africa and *Equus zebra hartmannae* of Namibia and Angola); and the Asian wild asses *Equus kiang* and *Equus hemionus* (with subspecies *Equus hemionus kulan* and *Equus hemionus onager*) [2,3]. The domestic donkey *E. africanus* is accepted as a subspecies of the African wild ass [4,5].

Geographically isolated donkey populations, referred to as breeds here, including Martina Franca (MF) of the continental Puglia region, Ragusano (RG) of the island province of Sicily, Pantesco (PT) of the small island of Pantelleria (85 km^2^), and Catalonian (CT) of the Spanish Catalonian region, share similar phenotypes due to genetic inheritance/ecological constraints and the outcomes of biodiversity conservation programs. The natural history of the donkey is of interest because the desire to safeguard genetic biodiversity is growing [6] and because the three autochthonous Italian breeds are considered endangered, including MF with 160 females and 36 males (Associazione Nazionale Asino Martina Franca) and RG with 344 females and 30 males (Domestic Animals Diversity Information System), which are critically endangered, as is PT with 47 females and 15 males (Associazione Italiana Allevatori).

The phylogeny of the domestic donkey is not yet well understood. Historical files show that the Roman Empire dominated the Iberian Peninsula for ~613 years from 218 BC to 395 [7]. At that time, the MF donkey was commonly used for transport and military operations; thus, it is presumed that it was widely distributed across Spain [8].

Later, Spain dominated central Southern Italy for ~148 years from 1559–1707 BC, and CT was likely introduced during this time in Italy. However, phenotypic traits and anthropological documents are often insufficient to ascertain breed history, origin, and the occurrence of genetic exchange [9]. Instead, mitochondrial DNA (mtDNA) sequencing can determine intra- and inter-species historical, biogeographic, and phylogenetic relationships [10]. The extrachromosomal mitochondrial genome, unlike the nuclear one, is inherited only through the maternal lineage, is haploid, and does not undergo genetic recombination [11,12].

The application of clonal polymorphisms to study the genetics of domestic animals is valuable [8,10]. Variation in the D-loop region of mtDNA and the lack of recombination in mtDNA make it a highly informative tool for matrilineal studies, for determining intraspecies phylogenetic relationships, and for characterizing intrabreed variation [12,13,14,15,16]. mtDNA studies of dog breeds, which have greater phenotypic and working variability compared to the donkey, which is relatively uniform, have revealed genetic information on their domestication, evolution, and hereditary diseases [17,18].

mtDNA studies of equine breeds were used to investigate their origin [19,20,21,22,23,24,25,26] and to track breed migration and distribution by comparing the maternal lines in different populations [27,28]. The complete donkey mitochondrial genome sequence was essential to date the divergence from the horse between 8 and 10 MYA [29,30], which is earlier than paleontological data [24,31] and data from restriction endonuclease analysis [32].

Interestingly, two lineages of the domestic donkey were identified using mtDNA: Clade 1 for the Nubian lineage (*E. a. africanus*) and Clade 2 for the Somali lineage (*E. a. somaliensis*). These lineages resulted from two separate domestication events among two wild ancestral populations located in (1) the Atbara region and Red Sea Hills (NW Sudan) and in (2) southern Eritrea, Ethiopia, and Somalia [5,33,34,35,36]. However, the existence of another ancestor of the domestic donkey belonging to an unrecognized extinct African wild population has been suggested [6,36,37].

Genetic studies on the biodiversity of the Italian donkey are limited and have primarily focused on variability among protein markers and microsatellites [38,39,40]. Recently, whole genome sequencing [41] and mtDNA [42] were used to study the evolution and genetic diversity of Italian donkey populations.

In this study, we evaluated the mtDNA D-loops of endangered and critically endangered Italian donkey breeds. mtDNA sequences, single-nucleotide polymorphisms (SNPs), and haplotypes were identified and analyzed to investigate the matrilineal assortment within and between asinine breeds with similar phenotypes and to investigate the origin and phylogenetic relationships between asinine breeds to better manage rare donkey breeds by establishing proper breeding and conservation programs.

## 2. Materials and Methods

One-hundred and twenty-three salivary samples were collected from eight official breeding stations (Appendix A additional material; Istituto Incremento Ippico associated with Facoltà di Medicina Veterinaria Università degli Studi di Teramo (Fondo Rustico Chiareto) and Centro di Conservazione del Patrimonio Genetico dell’Asino della razza Martina Franca (Azienda agricola Russoli Crispiano); Istituto Incremento Ippico Regione Campania Santa Maria Capua Vetere and Azienda Agricola Ciro Schirò Corleone-Monreale unfortunately have no successful sequenced samples) in accordance with the standards for care and protection of animals used for scientific purposes Directive 2010/63/EU. This study was approved by the Ethics Committee (Protocol No. 62128 of 27 April 2018). The samples collected were from free-range animals with certificates of origin, which were used to exclude animals of the same maternal descent, in order to increase genetic variability in the sample set. Seventy-seven samples, including MF = 27, RG = 22, PT = 8, CT = 19, and 1 Italian crossbreed, were sequenced successfully (Figure 1), whereas the remaining 46 samples were corrupted.

Genetic material was collected from saliva using a sterile oral swab, transferred to FTA mini-cards, and stored in multibarrier pouches (Whatman Labware Products, U.K.). The reference material is available at the O.V.U.D. (University Veterinary Hospital) Centre for the breeding of large animals at the Faculty of Veterinary Medicine, University of Teramo, Italy. Based on the complete donkey mtDNA sequence (GenBank X97337) [29], two pairs of primers were designed to amplify the hypervariable region between sites 15390 and 15750 [43], which is a fragment of the D-loop mtDNA (http://bioinfo.ut.ee/primer3, accessed on 4 April 2019). After extraction from the FTA mini-card, DNA was amplified by PCR in a 25 μL reaction containing 50 ng of DNA, 2.5 mM MgCl2, 0.2 mM each dNTP, 0.5 μM PER 5′- CC AAG GAC TAT CAA GGA AG-3′ and FOR 5′-TTG GAG GGA TTG CTG ATT TC-3′ primers, 1× PCR buffer, and 1 U of Taq DNA polymerase (Fermentas, Thermo Fischer Scientific). The amplification was performed using the Mastercycler thermal cycler (Eppendorf, USA) with the following conditions: initial denaturation at 94 °C for 5 min followed by 35 cycles of 94 °C for 30 s, 58 °C for 30 s, 72 °C for 30 s, and then, a final extension at 72 °C for 5 min.

The raw sequence trace files were checked for the presence of ambiguous bases using the software Chromas v.2.5.1 (http://www.technelysium.com.au/, accessed on 4 April 2019). Sequences were aligned with Muscle; in Appendix A is the alignment with other Italian donkey breeds [42]. The number of polymorphic sites (parsimony informative and singleton sites), the number of haplotypes (private and shared haplotypes), nucleotide diversity, and the average number of nucleotide differences were calculated according to Tajima (1983) and Nei (1987) using MEGA7, Fu’s neutrality statistic test, and Tajima’s D test with DnaSP 6.12.01, as well as using a maximum parsimony analysis and the maximum composite likelihood method. The median-joining network (for sequences, see Appendix A) and principal coordinates analysis (PCoA) were performed with DARwin software [44,45,46]; Appendix A were created by using itol.embl.de/tree [47]. The other statistical analyses were performed with Statistica 7.0 StatSoft.

## 3. Results

### 3.1. Breed Haplotype Analysis

The successfully analyzed samples from the eight certified breeding centers (Appendix A) included MF, RG, PT, and CT donkeys. The mtDNA D-loop hypervariable region between sites 15390 and 15750 (GenBank ID # 2466755, Appendix A) was fully sequenced for 77 samples, and 56 haplotypes including 33 polymorphic sites were found. Of these, 14.6% of haplotypes have frequencies greater than 2.7%: Hap 1, 22 and 36; Hap 30, 37, 40 and 53 F = 4.1%; Hap 51 F = 12.2% the most common; while 85.4% of haplotypes are rare and have frequencies of 1.4%. Table 1 shows molecular diversity indices for each breed. The breed genetic diversity for all the breeds, evaluated by the nucleotide diversity value (π), was 0.128. Within subpopulations, π was 0.098, and the mean interpopulation evolutionary π was 0.03. The haplotypes identified in the analyzed breeds included 6 for PT, 14 for CT, and 22 each for MF and RG.

All the sequenced samples were aligned with a reference sequence (GenBank X97337) to highlight the presence of SNPs. The absolute number of mutations identified for each sequenced sample is shown (Figure 2A, Table 2 and Appendix A). Moreover, all identified SNPs were classified according to their positions in the reference sequence and were characterized as transitions (purine–purine and pyrimidine–pyrimidine substitutions) or deletion (Figure 2B, left panel; every column represents a sequenced sample). The percentages of samples in which each SNP was identified are shown as histograms (Figure 2B, right panels, Table 2 and Appendix A).

The multivariate test of significance for nucleotide frequencies (Appendix A) showed no differences in nucleotide composition among breeds (*p* = 0.98). Maximum composite likelihood estimates of the nucleotide substitution pattern per breed, positions containing gaps, and missing data were eliminated as recommended by the literature (Appendix A) [45].

#### 3.1.1. Population Analyses

Higher base composition differences within and between breed sequences were found for MF and RG (Table 3).

Despite the limited number of samples, the following haplotypes were found: 17 in MF (Hap 1, 3, 5, 9, 10, 11, 15, 20, 24, 25, 34, 39, 41, 46, 47, 50, 54); 19 in RG (Hap 4, 6, 8, 12, 13, 16, 17, 18, 19, 21, 23, 29, 38, 42, 43, 44, 48, 49, 55); 9 in CT (Hap 2, 14, 26, 27, 28, 31, 33, 45, 52); and 3 in PT (Hap 7, 32, 35). The most represented seven haplotypes were Hap 51, common to all breeds; Hap 22, 36, and 37 found in MF and shared with PT and/or RG; Hap 30, distinctive of MF and also found in CT; and Hap 40 and 53, characteristic of CT.

The sequences were aligned with the reference sequence GenBank X97337 and other similar GenBank sequences representative of the Somali and Nubian African donkey lines, Chinese and other Asiatic *E. kiang*, *E. hemionus*, and *E. h. kulan* lines (Appendix A), and other Italian breeds (Appendix A as in Ref. [42]: Romagnolo donkey (ROD), Amiata donkey (AMD), Sardinian donkey (SAD), Asinara donkey (ASD), Ragusano donkey (RAD)).

#### 3.1.2. Origin and Phylogenetic Relationships

The phylogenetic relationships among the 55 haplotypes were calculated using a median-joining network. The haplotypes of each breed are color coded, the abundance of the haplotype indicated by the relative size of the symbol, and diffusion among the breeds is shown in color-coded pie charts (Figure 3). Looking at the distribution of haplotypes by breed, most of them are represented alone. In particular, MF and RG diverge from the reference sequence X97337 hap and show the highest variability. Conversely, PT is closely related and even shares haps with MF. Most CTs are grouped into two clusters: CT hap outgroups are closely related mainly to MF and completely share two haps with MF. Hap 51 is the most common and is shared among all races. The Fu neutrality statistic test and Tajima’s D test were performed to address the hypothesis of population expansion, and the results were Fs = −2.523, *p* < 0.05, and D = 2.004, *p* < 0.05, respectively.

PCoA analysis based on the dissimilarity matrix returned two different clusters, clusters I and II. Interestingly, in cluster II, there are only MF (6, 17, 27, 35) and RG (9, 21, 35, 53), while the rest of the haplotypes are grouped into cluster I. However, six haplotypes are not included in clusters I and II: they are MF Hap 22, 37, and 40; and RG Hap 13, 16, and 38 (Figure 4).

## 4. Discussion

### 4.1. Breed Molecular Analysis

Donkey breeds represent a fascinating model of domesticated biodiversity; thus, a number of studies have analyzed donkey pedigree and genetics. Pedigree and reconstruction studies usually lack the corresponding genetics [1,48,49], and genetic studies have frequently lacked lineage information [39,42,50].

Pedigree studies on Italian (MF, Amiata) and Spanish (CT, Andalusian, Miranda) donkeys found dramatic losses in genetic variation due to high rates of inbreeding [1,48,49,51]. However, pedigree incompleteness and the occurrence of a bottleneck event (e.g., MF in 1980) may have led to over- or under-estimations of genetic variation, which could affect breeding strategies [1,51]. To overcome this bias, in this study, genetic analysis was performed on subjects with certificates of origin from official authorized breeding centers. This approach allowed us to link mtDNA data to pedigree record, population, and breed to identify the same maternal descendants and to select individuals with presumed higher genetic variability to preserve biodiversity. A further bias in breed studies comes from unbalanced sample size comparisons for populations (e.g., difference of five-fold [39,42]), genetic structure, genetic variability, genetic robustness, average relatedness, inbreeding, co-ancestry, the degree of nonrandom mating, and origin [50,52]. In this study, we collected a similar number of samples per breed, except PT because of its limited number of breed lines; consequently, we analyzed a balanced sample. We found higher genetic variability in MF and RG, which disagrees with studies based on pedigree [1]. RG and MF are widely used on farms where natural random mating still occurs, and we believe that is the source of the variability. We found lower variability in PT, as expected, because of the limited number of individuals (*n* = 62) and because of its isolation. In the PT certificates of origin, eight distinct maternal lineages are attested, which is in agreement with the statement that the recovery of the PT breed started from a small nucleus of founders [39]. Furthermore, based on the number of haplotypes, the genetic robustness of PT is dramatically lower and the pedigree certificate alone cannot predict it. The low variability found in CT was not predicted because of the number of individuals in the population and the common distribution in the large region of Catalonia. This phenomenon could be the result of a breeding program with an unbalanced number of males vs. females (517 vs. 310 respectively; Asociacio del Foment dela Raca Asina Catala), rather than the result of the number of subjects and area occupied. Consequently, a potential bottleneck was produced by human artificial selection, which led to a loss in variability [42,49,53]. Because variability, robustness, and the degree of nonrandom mating decrease as average relatedness increases due to inbreeding and co-ancestry, a new reproduction program with multiple approaches is needed.

The molecular characteristics analyzed showed distinct nucleotide frequencies among breeds, which is in line with the literature [29,30]. The transition rate was greater for purines, and the transition rate per breed decreased from MF to RG to CT, which is partially discordant with other studies on the Italian donkey [42]. The nucleotide diversity (π) values are in line with a previous study on CT [54]. Cozzi et al. 2017 [42] found high nucleotide diversity in the MF donkey, but found low nucleotide diversity in the RG donkey.

We found the highest diversity in MF followed by RG and the lowest variability in CT followed by PT. The differences in our results and a previous study on the Italian donkey could be due to the following biases: (i) a lack of certificate of origin information leading to uncertain breed origin, (ii) unbalanced sampling between breeds with 74% of the breeds being Asinara and Sardo donkeys, both from the island of Sardinia [42]. Overall, molecular indices show greater genetic variability among the Italian than the Spanish donkeys according to the previous literature [42,49,53].

### 4.2. Population Analyses

The population analyses showed more differences within RG, followed by MF, CT, and PT. The RG genetic population structure is analogous to the highly heterogeneous large maternal Balkan donkey population with a more complex genetic structure than previously thought [53]. The Balkan donkey population is highly genetically diverse despite their severe population decline, probably due to introgression of other related breeds [53]. Introgression probably occurred for RG, but not MF.

Our analysis of PT is consistent with a microsatellite genetic variability study that showed lower variability in PT compared to RG and another Sicilian breed, the Grigio Siciliano (GS) [39]. The genetic variability observed in PT, RG, and GS [39] is lower than that reported in five Spanish breeds [54] and three Croatian breeds [55], but higher than that observed in the Amiata donkey from Italy [31] and in Chinese breeds [36]. The comparisons between breeds showed more differences between MF and RG and fewer differences between PT and CT, between RG and PT, and between RG and CT. The fewest differences are between PT and CT.

Two different studies on the Balkan donkey revealed different interpretations. In the first study, no correspondence between geographical areas and maternal genetic structure was found. Because the difference between the Balkan donkey and the African Burkina Faso donkey outgroup was also low, the authors could not trace routes of expansion in the donkey; consequently, they suggested that the species spread very quickly after domestication [56]. The second study assessed three Balkan donkey populations: Istrian (IS), north Adriatic (NA), and Littoral-Dinaric (LD), and their results suggested that IS is a unique breed, which mixed with LD during sporadic migration events, and that NA and LD are genetically similar [55]. Our study suggests similar effects of migration by MF on CT and PT, which are in accordance with historical reconstructions.

### 4.3. Origin and Phylogenetic Relationships

The well-established identification of two main lineages and the probable existence of another unrecognized extinct wild ancestor in domestic donkeys are believed to be the result of separate domestication events [5,6,33,34,35,36,37]. However, the genetic structure of the Chinese donkey indicates another possible line [36]. Therefore, in the donkey, such as in the dog [17], genetic data support multicentric breed origins. In agreement with this emerging theory is the identification of a potential new clade unique from the MF and RG Italian donkeys. Furthermore, another recent study also suggested multiple breed origins [42]. In Croatian and Serbian donkeys, three haplotype groups were found [55] with distinct nuclear gene pools [53]. A heterogeneous genetic structure of the Balkan donkey was hypothesized because there was no geographical structure; thus, it was difficult to trace the routes of expansion in the donkey [56]. However, other hypotheses for the complex genetic relationships among Italian donkey breeds and breeds living in the Mediterranean and Balkan areas [42] include ancestry and the genetic makeup of modern donkey populations. Our analysis suggests a multicentric domestication phenomena coupled with multiple waves of colonization and counter colonization, such as what occurred when the Roman Empire brought the MF donkey to Spain and when Spain brought the MF donkey to Italy; this is in accordance with the hypothesis suggested by Stanisic and colleagues [53].

There is a rising interest in maintaining genetic diversity in animal populations to safeguard the widest possible genetic resources through conservation programs [57,58].

For conservation of domestic breeds, the preservation of genetic capital is crucial because they are already zootechnical forms with reduced original natural variability. The biodiversity of the domestic form is the result of a genetic pool derived from interaction with semi-artificial environments regulated by human needs and human migratory movements. As a result, genetic and phenotypic changes with respect to the wild species of origin have been addressed slowly based on human necessity.

Donkey conservation represents a biological problem with regard to analogous phenotypes. The similar donkey breed phenotypes may be a result of genetic exchange among breeds, identical origin from an African or Asian population, equivalent climate conditions, and/or similar types of work carried out by the donkeys.

Our molecular analyses showed a dramatic loss in variability across all breeds tested, but especially in PT and CT. MF and RG have higher numbers of haplotypes and SNPs than PT and CT. In practice, the MF donkey is an important reservoir of biodiversity that must be preserved with the widest possible range of its genetic heritage, and errors of consanguinity for aesthetic purposes should be avoided with the help of conservation programs. Future conservation programs should include certificates of origin and genetic analyses of the matrilines at least.

## 5. Conclusions

In conclusion, this study: (i) identified D-loop mtDNA characteristics for the MF donkey and three phenotypically similar breeds, (ii) identified different matrilines in the MF donkey and similar breeds, (iii) identified the biodiversity of each breed, and (iv) determined the phylogenetic relationships among the breeds. This extensive study on the biodiversity and phylogenetic relationships of MF, RG, PT, and CT donkey breeds is useful for future domestication studies.

In this study, we analyzed the genetics of a limited number of endangered and highly endangered donkey breeds. The data showed significant loss in variability among all the breeds evaluated, which is in agreement with previous studies that used different methods. The primeval haplotypes, haplogroup variability, and large number of maternal lineages are preserved in the MF and RG breeds; thus, they play putative pivotal roles in the phyletic relationships of the studied donkey breeds. Given the level of endangerment undergone by these breeds, actions are necessary to ensure their long-term survival and conservation. Improving the reproduction and management of existing populations, clarifying their historic interactions by studying their population genetics, and extending and improved monitoring of maternal lineages are valid options.

## Figures and Tables

**Figure 1 genes-12-01109-f001:**
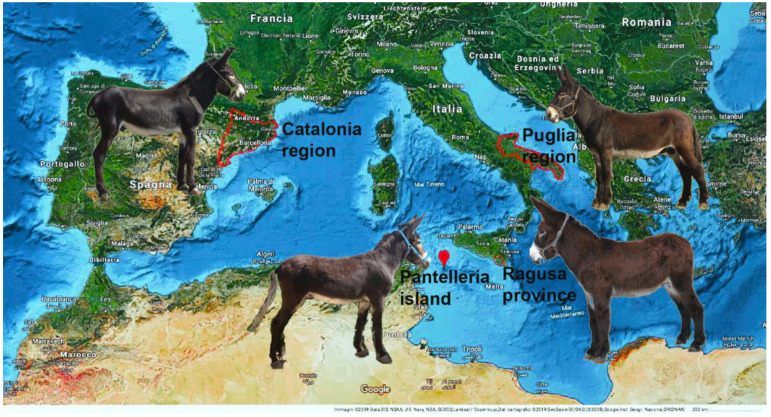
Breed distribution. The red perimeters pinpoint the original Mediterranean distribution of the four donkey breeds investigated: Martina Franca (MF), Ragusano (RG), Pantesco (PT), and Catalonian (CT).

**Figure 2 genes-12-01109-f002:**
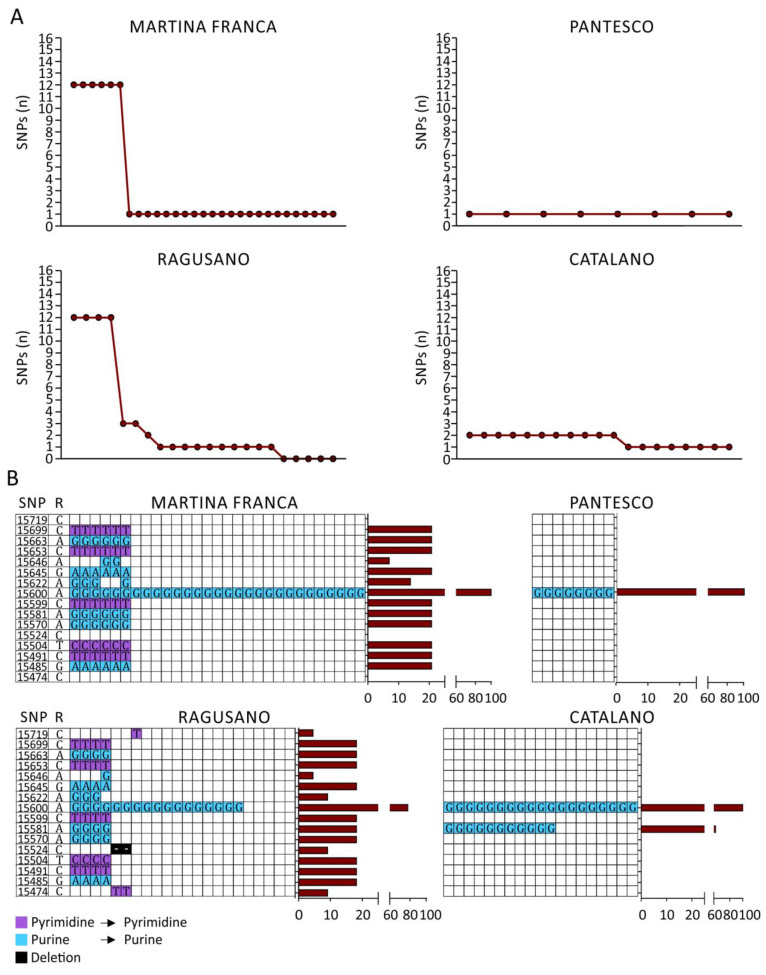
SNP analysis performed on mtDNA isolated from Martina Franca (MF), Ragusano (RG), Pantesco (PT), and Catalonian (CT) donkey breeds. (**A**) Absolute number of mutations found in each sample by Sanger sequencing. (**B**) Every SNP was characterized according to the position on the reference sequence, the kind of mutation, and the percentage of samples containing the mutation (brown histograms) using MEGA7. R = reference sequence (X97337).

**Figure 3 genes-12-01109-f003:**
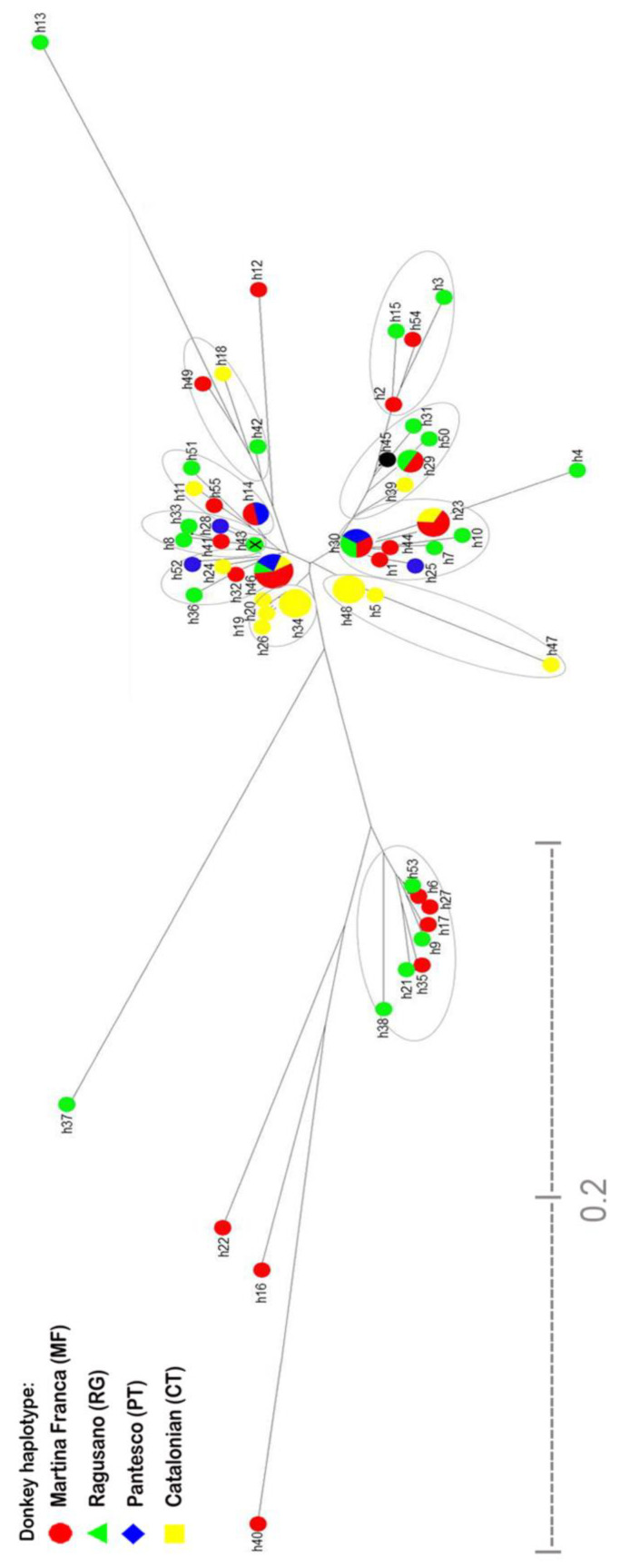
Median-joining network analysis. This analysis was based on 350 bp of the mtDNA D-loop sequences of 77 donkeys (Martina Franca (MF), Ragusano (RG), Pantesco (PT), Catalonian (CT)), which consists of 55 haplotypes (DARwin 6.0). Each breed is color coded, and for each haplotype, the proportions of the different breeds are shown, in black the crossbreed haplotype. The reference sequence X97337 is indicated as x.

**Figure 4 genes-12-01109-f004:**
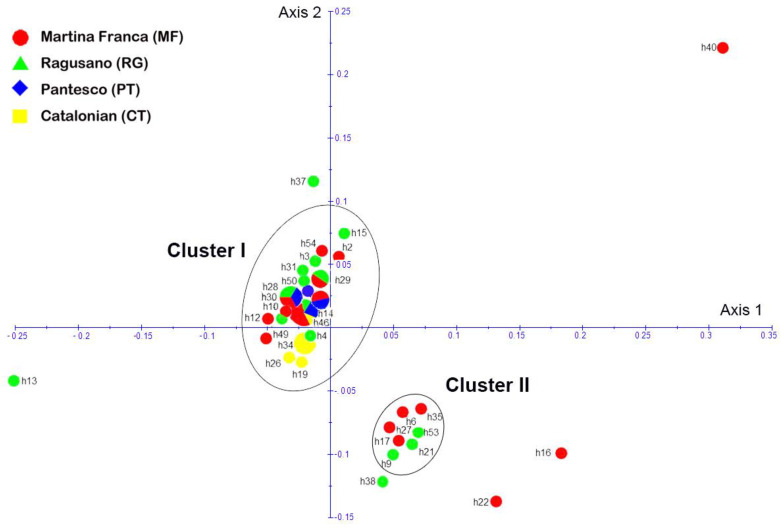
PCoA. The two axes of the PCoA plot were based on the dissimilarity matrix according to Kimura (1980) and generated using DARwin 6.0. Two different clusters, cluster I and cluster II, included most haplotypes of the breeds studied (Martina Franca (MF), Ragusano (RG), Pantesco (PT), Catalonian (CT)). Haplotypes that did not fall into either cluster were MF Hap 22, 37, and 40 and RG Hap 13, 16, and 38.

**Table 1 genes-12-01109-t001:** D-loop nucleotide polymorphisms and molecular diversity indices per breed tested in the study.

Breed	*n*	NHap	SNPs	π
Martina Franca (MF)	27	22(5 s)	13	0.162 ± 0.022
Ragusano (RG)	22	22(3 s)	15	0.132 ± 0.028
Pantesco (PT)	8	6(3 s)	1	0.025 ± 0.001
Catalonian (CT)	19	13(4 s)	2	0.038 ± 0.009
crossbreed	1	1	1	
ALL	77	56	33	0.128

s, shared haplotypes. In this dataset, there is a crossbreed with its own haplotype. *n*, number of individuals sampled per breed; NHap, the number of haplotypes in each breed with the number of shared haplotypes in parentheses; SNPs, the number of polymorphic sites; π, nucleotide diversity with standard deviation.

**Table 2 genes-12-01109-t002:** Absolute and relative haplotype frequencies in the four analyzed donkey breeds.

SNP			Martina Franca (MF)		Catalano (CT)		Pantesco (PT)		Ragusano (RG)	
Position	Reference	Mutation	Samples (*n*)	%	Samples (*n*)	%	Samples (*n*)	%	Samples (*n*)	%
15719	C	T	0	0	0	0	0	0	1	5
15699	C	T	6	21	0	0	0	0	4	18
15663	A	G	6	21	0	0	0	0	4	18
15653	C	T	6	21	0	0	0	0	4	18
15646	A	G	2	7	0	0	0	0	1	5
15645	G	A	6	21	0	0	0	0	4	18
15622	A	G	4	14	0	0	0	0	2	9
15600	A	G	29	100	19	100	8	100	17	77
15599	C	T	6	21	0	0	0	0	4	18
15581	A	G	6	21	11	58	0	0	4	18
15570	A	G	6	21	0	0	0	0	4	18
15524	C	DEL	0	0	0	0	0	0	2	9
15504	T	C	6	21	0	0	0	0	4	18
15491	C	T	6	21	0	0	0	0	4	18
15485	G	A	6	21	0	0	0	0	4	18
15474	C	T	0	0	0	0	0	0	2	9

**Table 3 genes-12-01109-t003:** Estimates of base composition differences within and between breed sequences using MEGA7.

**Breed**	**Mean within Groups**
Martina Franca (MF)	0.289
Ragusano (RG)	0.521
Pantesco (PT)	0.044
Catalonian (CT)	0.020
	**Mean between Groups**
MF × CT	0.162
MF × RG	0.387
MF × PT	0.158
RG × CT	0.276
RG × PT	0.278
CT × PT	0.031

## Data Availability

Sequences deposited in GenBank Submission No. 2466755 and showed in Appendix A.

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
