# Peer review of "Maternal Phylogenetic Relationships and Genetic Variation among Rare, Phenotypically Similar Donkey Breeds"

_genes, 2021, doi:10.3390/genes12081109_

Round 1

Reviewer 1 Report

The authors performed the necessary correction according most of my suggestions.

Please, check.

L24: you mean, 2.7%, 4.1% and 12.2% or 0.027, 0.041 and 0.122? Please, use the correct units to indicate the frequencies. Please, check also L150-151. 

Author Response

Comments and Suggestions for Authors

The authors performed the necessary correction according most of my suggestions.

Thank you very much for your comment.

Please, check.

L24: you mean, 2.7%, 4.1% and 12.2% or 0.027, 0.041 and 0.122? Please, use the correct units to indicate the frequencies. Please, check also L150-151. 

Answer: ok, corrections have been made

Reviewer 2 Report

The authors have much improved the manuscript, but I still have some remarks.

Table 2: thanks for add this table, but it is very difficult to identify the haplotypes. The table of haplotypes reported in most of papers allow the reader to view immediately the differences among the haplotypes (see the tables in Dell et al 2020 https://journals.plos.org/plosone/article?id=10.1371/journal.pone.0243247; Ivankovic et al 2009 https://www.agriculturejournals.cz/publicFiles/04939.pdf; Aranguren-Mendez et al 2004 https://www.agriculturejournals.cz/publicFiles/04939.pdf). Furthermore, I think the numbering of nucleotide positions have to be checked carefully.

The Authors added the sequences submitted to GenBank and I noticed that they are of different length and that there are many "N". Have you trimmed and correct the sequences before the alignment?

Supplementary materials: the CLUSTAL W alignment is very difficult to read. I think is better a table, similar to those suggested previously, that reported the nucleotide differences.

Please improve the quality of the Supplementary figures.

Author Response

Comments and Suggestions for Authors

The authors have much improved the manuscript, but I still have some remarks.

Thank you very much for your comment.

  1. Table 2: thanks for add this table, but it is very difficult to identify the haplotypes. The table of haplotypes reported in most of papers allow the reader to view immediately the differences among the haplotypes (see the tables in Dell et al 2020 https://journals.plos.org/plosone/article?id=10.1371/journal.pone.0243247; Ivankovic et al 2009 https://www.agriculturejournals.cz/publicFiles/04939.pdf; Aranguren-Mendez et al 2004 https://www.agriculturejournals.cz/publicFiles/04939.pdf). Furthermore, I think the numbering of nucleotide positions have to be checked carefully.

Answer: correction have been made.

  1. The Authors added the sequences submitted to GenBank and I noticed that they are of different length and that there are many "N". Have you trimmed and correct the sequences before the alignment?

Answer: Normally, we submit the original raw data obtained from Sanger sequencing to the GeneBank Repository in order to respect maximum transparency according to the Data Repository philosophy. Sequences have been trimmed for tails as normal procedure, similarly for N.

  1. Supplementary materials: the CLUSTAL W alignment is very difficult to read. I think is better a table, similar to those suggested previously, that reported the nucleotide differences.

Answer: In accordance with the referee's request we modified the table, and in order to leave the data clearly legible we adopted a colour code and (*) to easily read the sequence homology.

  1. Please improve the quality of the Supplementary figures.

Answer: done.

The figure are TIFF 300 DPI as requested by the journal. We upload the figure both within the word file and alone, since the quality is better, the Editor and Editorial Office could choose what is better.

Round 2

Reviewer 2 Report

 The manuscript has been significantly improved and no other revision
are needed.

This manuscript is a resubmission of an earlier submission. The following is a list of the peer review reports and author responses from that submission.

Round 1

Reviewer 1 Report

The study represents an example of single-marker phylogeographic study of animal populations (breeds here). The paper must be seriously revised due to the not clear presentation of results and following fundamental problems:

1)The authors use only a single short mitochondrial marker with low level of polymorphism. Sample size is very small. All the conclusions should take into account these facts.

2) I do not understand why UPGMA tree was used. There should be clear rationality behind or other method used (ex NJ) with bootstrap tasting and collapsing of the branches with low bootstrap support.

3) The manuscript should be shorten.

Author Response

Comments and Suggestions for Authors

The study represents an example of single-marker phylogeographic study of animal populations (breeds here). The paper must be seriously revised due to the not clear presentation of results and following fundamental problems:

  • The authors use only a single short mitochondrial marker with low level of polymorphism. Sample size is very small. All the conclusions should take into account these facts.

Line 67- We have used the short mitochondrial marker because that was described by Aranguren-Mendez J., Beja-Pereira A., Avellanet R., Dzama K., Jordana J. Mitochondrial DNA variation and genetic relationships in Spanish donkey breeds (Equus asinus) J. Anim. Breed. Genet. 2004;121:319–330. doi: 10.1111/j.1439-0388.2004.00464.x, and to compare with it. the number animal is limited, and especially the number of maternal lineage. However, we added a sentence in the conclusion that “genetic information arises from a limited number of endangered and highly endangered donkey breeds”.

2) I do not understand why UPGMA tree was used. There should be clear rationality behind or other method used (ex NJ) with bootstrap tasting and collapsing of the branches with low bootstrap support.

The UPGMA tree has been removed.

3- The manuscript should be shorten.

As requested by the reviewer the manuscript has been shortened.

Reviewer 2 Report

The present manuscript describes genetic diversity and relationships between donkey breeds based on analysis of D-loop mtDNA fragment. In total, 77 samples representing 4 donkey breeds were subjected to the study.

I believe, that the study will be interesting for the scientific community, but the extensive English editing should be performed before publishing.

Abstract

I believe, that the abstract should be rewritten. Please, begin with one or two sentences describing the importance of the conducted study. The aim of study must be formulated more specific. Please, provide the data concerning the number of animals within each of studied breeds. Please, describe shortly the materials and methods used for the study. The description of results should be also more specific. Please, confirm the main results by digit information. The conclusion should be based on research results.

Introduction

I suggest to start the introduction section from the describing the genus Equus (paragraph 2 of the introduction section in the present draft of the manuscript) and then follow by the description of donkey breeds (paragraph 1 of the introduction section in the manuscript).

L52: wa[8]s – please, correct

L58-61: probably, the part of text was lost; please, correct.

L70-71: “Further, the donkey complete mitochondrial genome sequence was greatly informative …”. What do you mean under this statement? Please, rephrase.  

Material and methods

L92 – In total, 123 samples were collected, but only 77 samples were sequenced. What about remaining 46 samples?

L 99 – I suggest to move table 1 in its present form to the supplemental materials. The data on haplotype distribution between studied breeds of donkeys in summarized form are presented in table 2 in Results section.

L109-110: please, indicate the primer positions in mtDNA reference sequence according to the gene bank accession number.  

Results

L126: I suppose, that the section 3.1 can be excluded. L127-132 can be moved to the section 3.2.1. I suggest to move Figure 1 to the introduction section, to the subsection, where the studied breeds were firstly introduced or to the materials and methods section. It is not necessary to duplicate the data concerning the number of haplotypes and number of polymorphic sites on Figure 1, because these data are presented in table 2.

L129: please, provide the exact length of D-loop fragment used for analysis. I suggest to indicate the first and last positions of this fragment according reference sequence.

L138: Molecular what?

L145: Table 2. Please, provide the standard deviations for nucleotide diversity values. According Tajima (1983), k is the average number of nucleotide differences (per DNA sequence) between any two sequences. I did not find the data concerning k values in table 2. What does it mean the values for T, C, A and G presented in the last four columns in table 2?

Additionally, I suggest to comprehensively evaluate the genetic diversity of the studies breeds by performing principal component analysis (PCA) on K, HapD and π using dimensionality reduction method.

I also suggest to test the hypothesis of population expansion calculating Fu’s neutrality statistic Fs and Tajima’s D test. You can use DnaSP 6.12.01 software.

L161-162, L167-170: I suggest to move table 3 and 4 to the supplemental materials.

Author Response

Comments and Suggestions for Authors

The present manuscript describes genetic diversity and relationships between donkey breeds based on analysis of D-loop mtDNA fragment. In total, 77 samples representing 4 donkey breeds were subjected to the study.

I believe, that the study will be interesting for the scientific community, but the extensive English editing should be performed before publishing.

We thank Rev. 2 for the positive comments and accordingly Rev. suggestions the manuscript has been revised by a professional English mother tongue company.

Abstract

I believe, that the abstract should be rewritten. Please, begin with one or two sentences describing the importance of the conducted study. The aim of study must be formulated more specific. Please, provide the data concerning the number of animals within each of studied breeds. Please, describe shortly the materials and methods used for the study. The description of results should be also more specific. Please, confirm the main results by digit information. The conclusion should be based on research results.

Abstract: we rewrote the abstract following Rev suggestions.

Introduction

I suggest to start the introduction section from the describing the genus Equus (paragraph 2 of the introduction section in the present draft of the manuscript) and then follow by the description of donkey breeds (paragraph 1 of the introduction section in the manuscript).

Thank you for this comment, the suggestion has been addressed.

L52: wa[8]s – please, correct

L58-61: probably, the part of text was lost; please, correct.

L70-71: “Further, the donkey complete mitochondrial genome sequence was greatly informative …”. What do you mean under this statement? Please, rephrase.  

Text has been modified concerning these lines.

Material and methods

L92 – In total, 123 samples were collected, but only 77 samples were sequenced. What about remaining 46 samples?

L 99 – I suggest to move table 1 in its present form to the supplemental materials. The data on haplotype distribution between studied breeds of donkeys in summarized form are presented in table 2 in Results section.

L109-110: please, indicate the primer positions in mtDNA reference sequence according to the gene bank accession number.  

As requested by the Rev we clarified and modified in the M&M the L92; L 99 and L109-110

Results

L126: I suppose, that the section 3.1 can be excluded. L127-132 can be moved to the section 3.2.1. I suggest to move Figure 1 to the introduction section, to the subsection, where the studied breeds were firstly introduced or to the materials and methods section. It is not necessary to duplicate the data concerning the number of haplotypes and number of polymorphic sites on Figure 1, because these data are presented in table 2.

L129: please, provide the exact length of D-loop fragment used for analysis. I suggest to indicate the first and last positions of this fragment according reference sequence.

The first and last positions have been added (hypervariable region between sites 15390 and 15750). as requested by the Rev.

L138: Molecular what?

Molecular has been deleted.

L145: Table 2. Please, provide the standard deviations for nucleotide diversity values. According Tajima (1983), k is the average number of nucleotide differences (per DNA sequence) between any two sequences. I did not find the data concerning k values in table 2. What does it mean the values for T, C, A and G presented in the last four columns in table 2?

As requested by the Rev the standard deviation has been provided. Table 2, now Tab 1, has been reformatted to be more clear.

Additionally, I suggest to comprehensively evaluate the genetic diversity of the studies breeds by performing principal component analysis (PCA) on K, HapD and π using dimensionality reduction method.

We apologize, were not able to address this suggestion.  

I also suggest to test the hypothesis of population expansion calculating Fu’s neutrality statistic Fs and Tajima’s D test. You can use DnaSP 6.12.01 software.

Line 183, pag.7. In order to test the hypothesis of population expansion was calculating Fu neutrality statistic test Fs -2,523, p < 0.05, and Tajima’s D test 2,004, p < 0.05.

L161-162, L167-170: I suggest to move table 3 and 4 to the supplemental materials.

Thank you for this suggestion, we moved the tables to supplemental material.

Reviewer 3 Report

This study examined the mtDNA variability in three Italian endangered donkey breeds (Martina Franca, Pantesco and Ragusano) and in Catalonian donkey. The objective are evaluate the genetic variability of mtDNA, asses the management of rare donkey breeds and establishing proper conservation program.

I expected to have a chance to read a straightforward and clear study, but in this case, I have to say I was a bit disappointed. There are many mistakes in grammar and in editing (Text and table). The quality of the figures have to be strongly improved. In the present layout is difficult to understand the meaning of some figures.

Still, I would like to encourage the authors to do some reanalyses, because the topic itself is interesting.

In addition to the problems in presentation, I have some major comments that require a bit more work from the authors.

1: the Authors chosen the samples based on certificate of origin, but in my experience, they must be carefully verify by a parentage testing, in order to avoid mistake on parents attribution.

2: the official mtDNA reference sequence for the donkey is the sequence X97337 (Xu et al. 1996). The Authors used this sequence for primers design, but compared the obtained sequence with a “reference mongrel samples” (Figures 3 and 4). I suggest replacing the mongrel sequence with the sequence X97337, which allows comparing the data of Italian donkey with those of other breeds.

3: the Authors should align the sequences of this study with those present in GenBank for the same breeds, in order to improve the information about the genetic diversity of the three Italian breeds and in Catalonian donkey.

More detailed comments are below:

Introduction

Line 52: change “wa [8] s”

Line 53: change “…(1559-17707 AC)” to “…(1559-17707 BC)”.

Line 88: change “…..phyletic” to “…phylogenetic”

Line 89: change “….in other”.

Materials and Methods

Table 1: this table have to be add as Supplementary Material. The footnotes of the table have to be insert in “Materials and Methods” section.

Line 106: add start and end bp position of the 350 bp fragment.

Line 109: change “...PER 5’” to “…FOR 5’”

Line 116: the sequences obtained have to be align with the reference sequence X97337.

Results

Please add and comment a table with polymorphic sites of the mtDNA sequences, according to the reference sequence X97337 and with absolute and relative haplotype frequencies.

Figure 1: please enlarge the figure. Legend: correct “….in vestigated”

Table 2: pay attention to the layout of the table! Please in the table and in all the paper, replace “mongrel” with “crossbred”.

Figure 2: please enlarge the figure. I do not understand the meaning of the analysis, a comment have to be add to the text.

Table 3, 4, and 5: please more comments of the tables are needed.

Table 5: the Authors presented the table as “Estimates of base composition bias difference between breed sequences”, but refers in the text as “The higher mean sequence distance within and between breed sequences are reported in Tab. 5”. Please clarify.

Line 191: are the haplotypes 55 or 56, as reported in “Results” section?

Discussion

Line 247: Cozzi et al 2017 reported the value for all the sequences. Please, reformulated the sentence.

Line 252: change “….biases in Cozzi et al [42]: . lack…..” to “…..biases in Cozzi et al [42]: i. lack ….”.

Line 250 to 255: I disagree with these sentences. Cozzi et al 2017 found a high nucleotide diversity and a high haplotype diversity in Martina Franca donkey, whereas in Ragusano donkey they found a low nucleotide diversity but a high haplotype diversity.

Line 255: “….. iii. the analysis does not distinguish among breeds” please clarify. In donkey as in other species, the haplotypes of mtDNA are shared and is impossible identify the breed by the haplotype.

Author Response

Comments and Suggestions for Authors

This study examined the mtDNA variability in three Italian endangered donkey breeds (Martina Franca, Pantesco and Ragusano) and in Catalonian donkey. The objective are evaluate the genetic variability of mtDNA, asses the management of rare donkey breeds and establishing proper conservation program.

I expected to have a chance to read a straightforward and clear study, but in this case, I have to say I was a bit disappointed. There are many mistakes in grammar and in editing (Text and table). The quality of the figures have to be strongly improved. In the present layout is difficult to understand the meaning of some figures.

Still, I would like to encourage the authors to do some reanalyses, because the topic itself is interesting.

In addition to the problems in presentation, I have some major comments that require a bit more work from the authors.

1: the Authors chosen the samples based on certificate of origin, but in my experience, they must be carefully verify by a parentage testing, in order to avoid mistake on parents attribution.

According to the REV., we used both strategies in order to verify if the certificate of origin are true in what they declare, this is useful for breeding programs that normally use certificate of origin. Infact, we already have written line 54-58 the following sentence: However, phenotypic traits and anthropological documents are often insufficient to ascertain the breed history, origin and occurrence of genetic exchange [9]. Instead, the mitochondrial DNA (mtDNA) properties simplify the understanding, through se-quencing, of historical, biogeographic and phylogenetic relationship in intra- and in-ter-species genetics structure [10].

2: the official mtDNA reference sequence for the donkey is the sequence X97337 (Xu et al. 1996). The Authors used this sequence for primers design, but compared the obtained sequence with a “reference mongrel samples” (Figures 3 and 4). I suggest replacing the mongrel sequence with the sequence X97337, which allows comparing the data of Italian donkey with those of other breeds.

A figure in the additional material has been added, with the following sentence:

In Figure A additional material all sequences were aligned with reference sequence GenBank X97337 and other similar sequences present in GenBank representative of African donkey lines Somalicus and Nubianus, Chinese ones and other Asiatic Equus kiang, E. hemionus and E. hemionus kulan.

And in the discussion:

Further, Chinese donkey genetic structure highlights another possible line [36], accordingly we have found this segregation in comparison with our data (Fig: A additional material), and we found the two African lines, Nubian and Somalicus.

3: the Authors should align the sequences of this study with those present in GenBank for the same breeds, in order to improve the information about the genetic diversity of the three Italian breeds and in Catalonian donkey.

The change has been done as requested by the REV, see point 2.

More detailed comments are below:

Introduction

Line 52: change “wa [8] s”

Corrections has been made.

Line 53: change “…(1559-17707 AC)” to “…(1559-17707 BC)”.

Correction made.

Line 88: change “…..phyletic” to “…phylogenetic”

Correction has been made.

Line 89: change “….in other”.

Correction has been made.

Materials and Methods

Table 1: this table have to be add as Supplementary Material. The footnotes of the table have to be insert in “Materials and Methods” section.

Table was moved to additional material.

Line 106: add start and end bp position of the 350 bp fragment.

Correction has been done

Line 109: change “...PER 5’” to “…FOR 5’”

Correction has been done

Line 116: the sequences obtained have to be align with the reference sequence X97337.

Correction has been done

Results

Please add and comment a table with polymorphic sites of the mtDNA sequences, according to the reference sequence X97337 and with absolute and relative haplotype frequencies.

Done as requested by the REV.

Figure 1: please enlarge the figure. Legend: correct “….in vestigated”

Correction has been made

Table 2: pay attention to the layout of the table! Please in the table and in all the paper, replace “mongrel” with “crossbred”.

Corrections have been done.

Figure 2: please enlarge the figure. I do not understand the meaning of the analysis, a comment have to be add to the text.

Correction done.

Table 3, 4, and 5: please more comments of the tables are needed.

The following sentences have been added:

Line 156. The haplotypes identified in the analyze breeds are from 6 PT, 14 CT and 22 for both MF and RG.

Line 182 Despite the limited number of samples private haplotypes was found: 16 in MF (1, 3, 5, 9, 10, 11, 15, 20, 24, 25, 34, 39, 41, 46, 47, 54); 19 in RG (4, 6, 8, 12, 13, 16, 17, 18, 19, 21, 23, 29, 38, 42, 43, 44, 48, 49, 55); 9 in CT (2, 14, 26, 27, 28, 31, 33, 45, 52); 3 in PT (7, 32, 35). Conversely, the most represented haplotypes are seven: 51 is common in all breeds; 22, 36 and 37 found in MF are shared with PT and/or RG; 30 distinctive of MF is also found in CT; 40 and 53 are characteristics of CT.

For tab 5 see below

Table 5: the Authors presented the table as “Estimates of base composition bias difference between breed sequences”, but refers in the text as “The higher mean sequence distance within and between breed sequences are reported in Tab. 5”. Please clarify.

The sentence has been corrected as requested by the REV.

Line 178 The higher base composition difference within and between breed sequences are for MF and RG as reported in Tab. 2.

Line 191: are the haplotypes 55 or 56, as reported in “Results” section?

Yes.

Discussion

Line 247: Cozzi et al 2017 reported the value for all the sequences. Please, reformulated the sentence.

Correction done.

Line 252: change “….biases in Cozzi et al [42]: . lack…..” to “…..biases in Cozzi et al [42]: i. lack ….”.

Correction done.

Line 250 to 255: I disagree with these sentences. Cozzi et al 2017 found a high nucleotide diversity and a high haplotype diversity in Martina Franca donkey, whereas in Ragusano donkey they found a low nucleotide diversity but a high haplotype diversity.

Thank you for this comment. Yes the REV is right, accordingly we reported entirely the sentence as suggested: “Cozzi et al 2017 found a high nucleotide diversity and a high haplotype diversity in Martina Franca donkey, whereas in Ragusano donkey they found a low nucleotide diversity but a high haplotype diversity.”

Line 255: “….. iii. the analysis does not distinguish among breeds” please clarify. In donkey as in other species, the haplotypes of mtDNA are shared and is impossible identify the breed by the haplotype.

We agree with the REV in biological terms of free ranging animals, but here we are in the same conditions of dog pure breed were official center perform let say ‘inbreeding’ between a given number of subject ascribable to a given breed; furthermore Pantesco is completely isolated from any other breed consequently as reported by Bannasch DL, Bannasch MJ, Ryun JR, Famula TR, Pedersen NC (2005) Y chromosome haplotype analysis in purebred dogs. Mamm Genome 16(4):273-80. doi: 10.1007/s00335-004-2435-8: “Breed-specific haplotypes were identified for 26 of the 50 breeds, and haplotype sharing between some breeds indicated a common history, with a significant genetic variation across breeds and with geographic origin of the breeds, particularly among breeds of African origin.”

Round 2

Reviewer 1 Report

The manuscript has been sufficiently improved.

Author Response

We thank the reviewer for appreciation of our manuscript.

Reviewer 3 Report

I have some major comments for the authors.

1: In the previous review, I asked a table with polymorphic sites of the mtDNA sequences, according to the reference sequence X97337 and with absolute and relative haplotype frequencies. In your new version I do not found the table that is a mandatory point.

2: Figure 2: the figure is really too small. I understand that the number refer to the SNPs identified, but they are not the real position on reference sequence X97337, so is impossible compare this figure with literature data. The figure have to be modify.

3: the Authors added sequence of African donkey Somalicus and Nubianus, Chinese ones and other Asiatic Equus kiang, E. hemionus and E. hemionus kulan, this is interesting but do not clarify the history of Italian donkeys. They should align the sequences of this study with those present in GenBank for the same breeds or for other Italian donkey breeds/populations, in order to improve the information about the genetic diversity of the Italian donkey breeds and in Catalonian donkey. I think this is mandatory and one of the goals of the paper.

4: The reference sequence for the median-joining network analysis should be X97337, not the crossbred donkey. Please make again the analyses with the correct reference.

5: please add the GenBank accession numbers for your sequences.

6: The sequences used for the NJ have to be listed in “Materials and Methods” section or added as Supplementary materials.

7: Please specify in “Materials and Methods” section the software used for the analyses in supplementary figure A.

Other comments:

Line 179-182: are you sure that your typical haplotypes are not present in other Italian donkey breeds? For this reason, I suggest the alignment with other sequences from Italian donkey from GenBank.

Line 189: please change “3.2.3. Origin and phyletic relationships” in “3.2.3. Origin and phylogenetic relationships”

Line 190: please change “The phyletic relationships…” in “The phylogenetic relationships…..”

Line 255: “….. iii. the analysis does not distinguish among breeds”.

We agree with the REV in biological terms of free ranging animals, but here we are in the same conditions of dog pure breed were official center perform let say ‘inbreeding’ between a given number of subject ascribable to a given breed; furthermore Pantesco is completely isolated from any other breed consequently as reported by Bannasch DL, Bannasch MJ, Ryun JR, Famula TR, Pedersen NC (2005) Y chromosome haplotype analysis in purebred dogs. Mamm Genome 16(4):273-80. doi: 10.1007/s00335-004-2435-8: “Breed-specific haplotypes were identified for 26 of the 50 breeds, and haplotype sharing between some breeds indicated a common history, with a significant genetic variation across breeds and with geographic origin of the breeds, particularly among breeds of African origin.”

Sorry, but donkey bred is quite different from dog bred. Dog pure breeds are mainly selected using inbreeding matings with a high level of consanguinity and the breed differentiation is high, as reported in many papers using different markers (for example Sechi et al., 2016; https://doi.org/10.1080/1828051X.2016.1248867; Yang et al., 2019 https://doi.org/10.3389/fgene.2019.01174; Plassais et al., 2019 https://doi.org/10.1038/s41467-019-09373-w). The selective pressure operated by breeders is not so high in donkeys as in horses, so the haplotypes of mtDNA are shared and is impossible identify the breed by the haplotype, as well reported by Stanisic et al 2020 for the Banat donkey (Stanisic et al., 2020; https://doi.org/10.7717/peerj.8598).

In my opinion, the sentence have to be removed.

Line 283: please change “phyletic” with “phylogenetic”.

Line 286-391: Please clarify because the circular NJ in figure A is unclear, and I do not found evidence of a Chinese line and two African lines, Nubian and Somali, in your data.

Line 317-318: This sentence is wrong and have to be removed. It is well known that, over time, PT and RG have repeatedly crossbred each other and with the MF or with the Catalan donkey.

Line 320-330: These sentences have to be move and match with “Conclusion” section or delete. Again, change “phyletic” with “phylogenetic”!

Author Response

We thank the reveiwer for the important contribution to the improvement of the article. All your concerns have been addressed.

1: In the previous review, I asked a table with polymorphic sites of the mtDNA sequences, according to the reference sequence X97337 and with absolute and relative haplotype frequencies. In your new version I do not found the table that is a mandatory point.

We apologize for this misunderstanding, according to the reviewer request, we added a table (new table 2) including absolute and relative haplotype frequencies.

2: Figure 2: the figure is really too small. I understand that the number refer to the SNPs identified, but they are not the real position on reference sequence X97337, so is impossible compare this figure with literature data. The figure have to be modify.

We thank the reviewer to highlight this point. We modified figure 2 in order to overcome these concerns: SNP numbers refer now to real position on the reference sequence. Plot and text were increased in size.

3: the Authors added sequence of African donkey Somalicus and Nubianus, Chinese ones and other Asiatic Equus kiang, E. hemionus and E. hemionus kulan, this is interesting but do not clarify the history of Italian donkeys. They should align the sequences of this study with those present in GenBank for the same breeds or for other Italian donkey breeds/populations, in order to improve the information about the genetic diversity of the Italian donkey breeds and in Catalonian donkey. I think this is mandatory and one of the goals of the paper.

We added the ClustalW alignment Supplementary materials with other Italian donkey, Tab B see line 133 and Fig B Italian alignment line 194

4: The reference sequence for the median-joining network analysis should be X97337, not the crossbred donkey. Please make again the analyses with the correct reference.

It was done using as reference X97337 which share the same haplotype with Ragusano donkey, now it has been clearly identify by x, see line 208. The reference sequence X97337 is indicated as x.

5: please add the GenBank accession numbers for your sequences.

We added in Supplementary materials the Submission ID # 2466755 Tab C see line 149

6: The sequences used for the NJ have to be listed in “Materials and Methods” section or added as Supplementary materials.

We added sequences in Supplementary materials Tab C see line 140.

7: Please specify in “Materials and Methods” section the software used for the analyses in supplementary figure A.

Supplementary material figure A and B were performed by using itol.embl.de/tree [Ci-tation: Letunic and Bork (2021) Nucleic Acids Res doi: 10.1093/nar/gkab301].

Other comments:

Line 179-182: are you sure that your typical haplotypes are not present in other Italian donkey breeds? For this reason, I suggest the alignment with other sequences from Italian donkey from GenBank.

Accordingly, we removed the world private and as already done, see major points, we perform alignment with GenBank

Line 189: please change “3.2.3. Origin and phyletic relationships” in “3.2.3. Origin and phylogenetic relationships”

done

Line 190: please change “The phyletic relationships…” in “The phylogenetic relationships…..”

done

Line 255: “….. iii. the analysis does not distinguish among breeds”.

It has been removed

We agree with the REV in biological terms of free ranging animals, but here we are in the same conditions of dog pure breed were official center perform let say ‘inbreeding’ between a given number of subject ascribable to a given breed; furthermore Pantesco is completely isolated from any other breed consequently as reported by Bannasch DL, Bannasch MJ, Ryun JR, Famula TR, Pedersen NC (2005) Y chromosome haplotype analysis in purebred dogs. Mamm Genome 16(4):273-80. doi: 10.1007/s00335-004-2435-8: “Breed-specific haplotypes were identified for 26 of the 50 breeds, and haplotype sharing between some breeds indicated a common history, with a significant genetic variation across breeds and with geographic origin of the breeds, particularly among breeds of African origin.”

Sorry, but donkey bred is quite different from dog bred. Dog pure breeds are mainly selected using inbreeding matings with a high level of consanguinity and the breed differentiation is high, as reported in many papers using different markers (for example Sechi et al., 2016; https://doi.org/10.1080/1828051X.2016.1248867; Yang et al., 2019 https://doi.org/10.3389/fgene.2019.01174; Plassais et al., 2019 https://doi.org/10.1038/s41467-019-09373-w). The selective pressure operated by breeders is not so high in donkeys as in horses, so the haplotypes of mtDNA are shared and is impossible identify the breed by the haplotype, as well reported by Stanisic et al 2020 for the Banat donkey (Stanisic et al., 2020; https://doi.org/10.7717/peerj.8598).

In my opinion, the sentence have to be removed.

Line 283: please change “phyletic” with “phylogenetic”.

done

Line 286-391: Please clarify because the circular NJ in figure A is unclear, and I do not found evidence of a Chinese line and two African lines, Nubian and Somali, in your data.

It has been removed.

Line 317-318: This sentence is wrong and have to be removed. It is well known that, over time, PT and RG have repeatedly crossbred each other and with the MF or with the Catalan donkey.

 It has been removed.

Line 320-330: These sentences have to be move and match with “Conclusion” section or delete. Again, change “phyletic” with “phylogenetic”!

It was moved in the conclusion and changed.

We thank the reveiwer for the important contribution to the improvement of the article. All your concerns have been addressed.

1: In the previous review, I asked a table with polymorphic sites of the mtDNA sequences, according to the reference sequence X97337 and with absolute and relative haplotype frequencies. In your new version I do not found the table that is a mandatory point.

We apologize for this misunderstanding, according to the reviewer request, we added a table (new table 2) including absolute and relative haplotype frequencies.

2: Figure 2: the figure is really too small. I understand that the number refer to the SNPs identified, but they are not the real position on reference sequence X97337, so is impossible compare this figure with literature data. The figure have to be modify.

We thank the reviewer to highlight this point. We modified figure 2 in order to overcome these concerns: SNP numbers refer now to real position on the reference sequence. Plot and text were increased in size.

3: the Authors added sequence of African donkey Somalicus and Nubianus, Chinese ones and other Asiatic Equus kiang, E. hemionus and E. hemionus kulan, this is interesting but do not clarify the history of Italian donkeys. They should align the sequences of this study with those present in GenBank for the same breeds or for other Italian donkey breeds/populations, in order to improve the information about the genetic diversity of the Italian donkey breeds and in Catalonian donkey. I think this is mandatory and one of the goals of the paper.

We added the ClustalW alignment Supplementary materials with other Italian donkey, Tab B see line 133 and Fig B Italian alignment line 194

4: The reference sequence for the median-joining network analysis should be X97337, not the crossbred donkey. Please make again the analyses with the correct reference.

It was done using as reference X97337 which share the same haplotype with Ragusano donkey, now it has been clearly identify by x, see line 208. The reference sequence X97337 is indicated as x.

5: please add the GenBank accession numbers for your sequences.

We added in Supplementary materials the Submission ID # 2466755 Tab C see line 149

6: The sequences used for the NJ have to be listed in “Materials and Methods” section or added as Supplementary materials.

We added sequences in Supplementary materials Tab C see line 140.

7: Please specify in “Materials and Methods” section the software used for the analyses in supplementary figure A.

Supplementary material figure A and B were performed by using itol.embl.de/tree [Ci-tation: Letunic and Bork (2021) Nucleic Acids Res doi: 10.1093/nar/gkab301].

Other comments:

Line 179-182: are you sure that your typical haplotypes are not present in other Italian donkey breeds? For this reason, I suggest the alignment with other sequences from Italian donkey from GenBank.

Accordingly, we removed the world private and as already done, see major points, we perform alignment with GenBank

Line 189: please change “3.2.3. Origin and phyletic relationships” in “3.2.3. Origin and phylogenetic relationships”

done

Line 190: please change “The phyletic relationships…” in “The phylogenetic relationships…..”

done

Line 255: “….. iii. the analysis does not distinguish among breeds”.

It has been removed

We agree with the REV in biological terms of free ranging animals, but here we are in the same conditions of dog pure breed were official center perform let say ‘inbreeding’ between a given number of subject ascribable to a given breed; furthermore Pantesco is completely isolated from any other breed consequently as reported by Bannasch DL, Bannasch MJ, Ryun JR, Famula TR, Pedersen NC (2005) Y chromosome haplotype analysis in purebred dogs. Mamm Genome 16(4):273-80. doi: 10.1007/s00335-004-2435-8: “Breed-specific haplotypes were identified for 26 of the 50 breeds, and haplotype sharing between some breeds indicated a common history, with a significant genetic variation across breeds and with geographic origin of the breeds, particularly among breeds of African origin.”

Sorry, but donkey bred is quite different from dog bred. Dog pure breeds are mainly selected using inbreeding matings with a high level of consanguinity and the breed differentiation is high, as reported in many papers using different markers (for example Sechi et al., 2016; https://doi.org/10.1080/1828051X.2016.1248867; Yang et al., 2019 https://doi.org/10.3389/fgene.2019.01174; Plassais et al., 2019 https://doi.org/10.1038/s41467-019-09373-w). The selective pressure operated by breeders is not so high in donkeys as in horses, so the haplotypes of mtDNA are shared and is impossible identify the breed by the haplotype, as well reported by Stanisic et al 2020 for the Banat donkey (Stanisic et al., 2020; https://doi.org/10.7717/peerj.8598).

In my opinion, the sentence have to be removed.

Line 283: please change “phyletic” with “phylogenetic”.

done

Line 286-391: Please clarify because the circular NJ in figure A is unclear, and I do not found evidence of a Chinese line and two African lines, Nubian and Somali, in your data.

It has been removed.

Line 317-318: This sentence is wrong and have to be removed. It is well known that, over time, PT and RG have repeatedly crossbred each other and with the MF or with the Catalan donkey.

 It has been removed.

Line 320-330: These sentences have to be move and match with “Conclusion” section or delete. Again, change “phyletic” with “phylogenetic”!

It was moved in the conclusion and changed.

We thank the reveiwer for the important contribution to the improvement of the article. All your concerns have been addressed.

1: In the previous review, I asked a table with polymorphic sites of the mtDNA sequences, according to the reference sequence X97337 and with absolute and relative haplotype frequencies. In your new version I do not found the table that is a mandatory point.

We apologize for this misunderstanding, according to the reviewer request, we added a table (new table 2) including absolute and relative haplotype frequencies.

2: Figure 2: the figure is really too small. I understand that the number refer to the SNPs identified, but they are not the real position on reference sequence X97337, so is impossible compare this figure with literature data. The figure have to be modify.

We thank the reviewer to highlight this point. We modified figure 2 in order to overcome these concerns: SNP numbers refer now to real position on the reference sequence. Plot and text were increased in size.

3: the Authors added sequence of African donkey Somalicus and Nubianus, Chinese ones and other Asiatic Equus kiang, E. hemionus and E. hemionus kulan, this is interesting but do not clarify the history of Italian donkeys. They should align the sequences of this study with those present in GenBank for the same breeds or for other Italian donkey breeds/populations, in order to improve the information about the genetic diversity of the Italian donkey breeds and in Catalonian donkey. I think this is mandatory and one of the goals of the paper.

We added the ClustalW alignment Supplementary materials with other Italian donkey, Tab B see line 133 and Fig B Italian alignment line 194

4: The reference sequence for the median-joining network analysis should be X97337, not the crossbred donkey. Please make again the analyses with the correct reference.

It was done using as reference X97337 which share the same haplotype with Ragusano donkey, now it has been clearly identify by x, see line 208. The reference sequence X97337 is indicated as x.

5: please add the GenBank accession numbers for your sequences.

We added in Supplementary materials the Submission ID # 2466755 Tab C see line 149

6: The sequences used for the NJ have to be listed in “Materials and Methods” section or added as Supplementary materials.

We added sequences in Supplementary materials Tab C see line 140.

7: Please specify in “Materials and Methods” section the software used for the analyses in supplementary figure A.

Supplementary material figure A and B were performed by using itol.embl.de/tree [Ci-tation: Letunic and Bork (2021) Nucleic Acids Res doi: 10.1093/nar/gkab301].

Other comments:

Line 179-182: are you sure that your typical haplotypes are not present in other Italian donkey breeds? For this reason, I suggest the alignment with other sequences from Italian donkey from GenBank.

Accordingly, we removed the world private and as already done, see major points, we perform alignment with GenBank

Line 189: please change “3.2.3. Origin and phyletic relationships” in “3.2.3. Origin and phylogenetic relationships”

done

Line 190: please change “The phyletic relationships…” in “The phylogenetic relationships…..”

done

Line 255: “….. iii. the analysis does not distinguish among breeds”.

It has been removed

We agree with the REV in biological terms of free ranging animals, but here we are in the same conditions of dog pure breed were official center perform let say ‘inbreeding’ between a given number of subject ascribable to a given breed; furthermore Pantesco is completely isolated from any other breed consequently as reported by Bannasch DL, Bannasch MJ, Ryun JR, Famula TR, Pedersen NC (2005) Y chromosome haplotype analysis in purebred dogs. Mamm Genome 16(4):273-80. doi: 10.1007/s00335-004-2435-8: “Breed-specific haplotypes were identified for 26 of the 50 breeds, and haplotype sharing between some breeds indicated a common history, with a significant genetic variation across breeds and with geographic origin of the breeds, particularly among breeds of African origin.”

Sorry, but donkey bred is quite different from dog bred. Dog pure breeds are mainly selected using inbreeding matings with a high level of consanguinity and the breed differentiation is high, as reported in many papers using different markers (for example Sechi et al., 2016; https://doi.org/10.1080/1828051X.2016.1248867; Yang et al., 2019 https://doi.org/10.3389/fgene.2019.01174; Plassais et al., 2019 https://doi.org/10.1038/s41467-019-09373-w). The selective pressure operated by breeders is not so high in donkeys as in horses, so the haplotypes of mtDNA are shared and is impossible identify the breed by the haplotype, as well reported by Stanisic et al 2020 for the Banat donkey (Stanisic et al., 2020; https://doi.org/10.7717/peerj.8598).

In my opinion, the sentence have to be removed.

Line 283: please change “phyletic” with “phylogenetic”.

done

Line 286-391: Please clarify because the circular NJ in figure A is unclear, and I do not found evidence of a Chinese line and two African lines, Nubian and Somali, in your data.

It has been removed.

Line 317-318: This sentence is wrong and have to be removed. It is well known that, over time, PT and RG have repeatedly crossbred each other and with the MF or with the Catalan donkey.

 It has been removed.

Line 320-330: These sentences have to be move and match with “Conclusion” section or delete. Again, change “phyletic” with “phylogenetic”!

It was moved in the conclusion and changed.

We thank the reveiwer for the important contribution to the improvement of the article. All your concerns have been addressed.

1: In the previous review, I asked a table with polymorphic sites of the mtDNA sequences, according to the reference sequence X97337 and with absolute and relative haplotype frequencies. In your new version I do not found the table that is a mandatory point.

We apologize for this misunderstanding, according to the reviewer request, we added a table (new table 2) including absolute and relative haplotype frequencies.

2: Figure 2: the figure is really too small. I understand that the number refer to the SNPs identified, but they are not the real position on reference sequence X97337, so is impossible compare this figure with literature data. The figure have to be modify.

We thank the reviewer to highlight this point. We modified figure 2 in order to overcome these concerns: SNP numbers refer now to real position on the reference sequence. Plot and text were increased in size.

3: the Authors added sequence of African donkey Somalicus and Nubianus, Chinese ones and other Asiatic Equus kiang, E. hemionus and E. hemionus kulan, this is interesting but do not clarify the history of Italian donkeys. They should align the sequences of this study with those present in GenBank for the same breeds or for other Italian donkey breeds/populations, in order to improve the information about the genetic diversity of the Italian donkey breeds and in Catalonian donkey. I think this is mandatory and one of the goals of the paper.

We added the ClustalW alignment Supplementary materials with other Italian donkey, Tab B see line 133 and Fig B Italian alignment line 194

4: The reference sequence for the median-joining network analysis should be X97337, not the crossbred donkey. Please make again the analyses with the correct reference.

It was done using as reference X97337 which share the same haplotype with Ragusano donkey, now it has been clearly identify by x, see line 208. The reference sequence X97337 is indicated as x.

5: please add the GenBank accession numbers for your sequences.

We added in Supplementary materials the Submission ID # 2466755 Tab C see line 149

6: The sequences used for the NJ have to be listed in “Materials and Methods” section or added as Supplementary materials.

We added sequences in Supplementary materials Tab C see line 140.

7: Please specify in “Materials and Methods” section the software used for the analyses in supplementary figure A.

Supplementary material figure A and B were performed by using itol.embl.de/tree [Ci-tation: Letunic and Bork (2021) Nucleic Acids Res doi: 10.1093/nar/gkab301].

Other comments:

Line 179-182: are you sure that your typical haplotypes are not present in other Italian donkey breeds? For this reason, I suggest the alignment with other sequences from Italian donkey from GenBank.

Accordingly, we removed the world private and as already done, see major points, we perform alignment with GenBank

Line 189: please change “3.2.3. Origin and phyletic relationships” in “3.2.3. Origin and phylogenetic relationships”

done

Line 190: please change “The phyletic relationships…” in “The phylogenetic relationships…..”

done

Line 255: “….. iii. the analysis does not distinguish among breeds”.

It has been removed

We agree with the REV in biological terms of free ranging animals, but here we are in the same conditions of dog pure breed were official center perform let say ‘inbreeding’ between a given number of subject ascribable to a given breed; furthermore Pantesco is completely isolated from any other breed consequently as reported by Bannasch DL, Bannasch MJ, Ryun JR, Famula TR, Pedersen NC (2005) Y chromosome haplotype analysis in purebred dogs. Mamm Genome 16(4):273-80. doi: 10.1007/s00335-004-2435-8: “Breed-specific haplotypes were identified for 26 of the 50 breeds, and haplotype sharing between some breeds indicated a common history, with a significant genetic variation across breeds and with geographic origin of the breeds, particularly among breeds of African origin.”

Sorry, but donkey bred is quite different from dog bred. Dog pure breeds are mainly selected using inbreeding matings with a high level of consanguinity and the breed differentiation is high, as reported in many papers using different markers (for example Sechi et al., 2016; https://doi.org/10.1080/1828051X.2016.1248867; Yang et al., 2019 https://doi.org/10.3389/fgene.2019.01174; Plassais et al., 2019 https://doi.org/10.1038/s41467-019-09373-w). The selective pressure operated by breeders is not so high in donkeys as in horses, so the haplotypes of mtDNA are shared and is impossible identify the breed by the haplotype, as well reported by Stanisic et al 2020 for the Banat donkey (Stanisic et al., 2020; https://doi.org/10.7717/peerj.8598).

In my opinion, the sentence have to be removed.

Line 283: please change “phyletic” with “phylogenetic”.

done

Line 286-391: Please clarify because the circular NJ in figure A is unclear, and I do not found evidence of a Chinese line and two African lines, Nubian and Somali, in your data.

It has been removed.

Line 317-318: This sentence is wrong and have to be removed. It is well known that, over time, PT and RG have repeatedly crossbred each other and with the MF or with the Catalan donkey.

 It has been removed.

Line 320-330: These sentences have to be move and match with “Conclusion” section or delete. Again, change “phyletic” with “phylogenetic”!

It was moved in the conclusion and changed.

We thank the reveiwer for the important contribution to the improvement of the article. All your concerns have been addressed.

1: In the previous review, I asked a table with polymorphic sites of the mtDNA sequences, according to the reference sequence X97337 and with absolute and relative haplotype frequencies. In your new version I do not found the table that is a mandatory point.

We apologize for this misunderstanding, according to the reviewer request, we added a table (new table 2) including absolute and relative haplotype frequencies.

2: Figure 2: the figure is really too small. I understand that the number refer to the SNPs identified, but they are not the real position on reference sequence X97337, so is impossible compare this figure with literature data. The figure have to be modify.

We thank the reviewer to highlight this point. We modified figure 2 in order to overcome these concerns: SNP numbers refer now to real position on the reference sequence. Plot and text were increased in size.

3: the Authors added sequence of African donkey Somalicus and Nubianus, Chinese ones and other Asiatic Equus kiang, E. hemionus and E. hemionus kulan, this is interesting but do not clarify the history of Italian donkeys. They should align the sequences of this study with those present in GenBank for the same breeds or for other Italian donkey breeds/populations, in order to improve the information about the genetic diversity of the Italian donkey breeds and in Catalonian donkey. I think this is mandatory and one of the goals of the paper.

We added the ClustalW alignment Supplementary materials with other Italian donkey, Tab B see line 133 and Fig B Italian alignment line 194

4: The reference sequence for the median-joining network analysis should be X97337, not the crossbred donkey. Please make again the analyses with the correct reference.

It was done using as reference X97337 which share the same haplotype with Ragusano donkey, now it has been clearly identify by x, see line 208. The reference sequence X97337 is indicated as x.

5: please add the GenBank accession numbers for your sequences.

We added in Supplementary materials the Submission ID # 2466755 Tab C see line 149

6: The sequences used for the NJ have to be listed in “Materials and Methods” section or added as Supplementary materials.

We added sequences in Supplementary materials Tab C see line 140.

7: Please specify in “Materials and Methods” section the software used for the analyses in supplementary figure A.

Supplementary material figure A and B were performed by using itol.embl.de/tree [Ci-tation: Letunic and Bork (2021) Nucleic Acids Res doi: 10.1093/nar/gkab301].

Other comments:

Line 179-182: are you sure that your typical haplotypes are not present in other Italian donkey breeds? For this reason, I suggest the alignment with other sequences from Italian donkey from GenBank.

Accordingly, we removed the world private and as already done, see major points, we perform alignment with GenBank

Line 189: please change “3.2.3. Origin and phyletic relationships” in “3.2.3. Origin and phylogenetic relationships”

done

Line 190: please change “The phyletic relationships…” in “The phylogenetic relationships…..”

done

Line 255: “….. iii. the analysis does not distinguish among breeds”.

It has been removed

We agree with the REV in biological terms of free ranging animals, but here we are in the same conditions of dog pure breed were official center perform let say ‘inbreeding’ between a given number of subject ascribable to a given breed; furthermore Pantesco is completely isolated from any other breed consequently as reported by Bannasch DL, Bannasch MJ, Ryun JR, Famula TR, Pedersen NC (2005) Y chromosome haplotype analysis in purebred dogs. Mamm Genome 16(4):273-80. doi: 10.1007/s00335-004-2435-8: “Breed-specific haplotypes were identified for 26 of the 50 breeds, and haplotype sharing between some breeds indicated a common history, with a significant genetic variation across breeds and with geographic origin of the breeds, particularly among breeds of African origin.”

Sorry, but donkey bred is quite different from dog bred. Dog pure breeds are mainly selected using inbreeding matings with a high level of consanguinity and the breed differentiation is high, as reported in many papers using different markers (for example Sechi et al., 2016; https://doi.org/10.1080/1828051X.2016.1248867; Yang et al., 2019 https://doi.org/10.3389/fgene.2019.01174; Plassais et al., 2019 https://doi.org/10.1038/s41467-019-09373-w). The selective pressure operated by breeders is not so high in donkeys as in horses, so the haplotypes of mtDNA are shared and is impossible identify the breed by the haplotype, as well reported by Stanisic et al 2020 for the Banat donkey (Stanisic et al., 2020; https://doi.org/10.7717/peerj.8598).

In my opinion, the sentence have to be removed.

Line 283: please change “phyletic” with “phylogenetic”.

done

Line 286-391: Please clarify because the circular NJ in figure A is unclear, and I do not found evidence of a Chinese line and two African lines, Nubian and Somali, in your data.

It has been removed.

Line 317-318: This sentence is wrong and have to be removed. It is well known that, over time, PT and RG have repeatedly crossbred each other and with the MF or with the Catalan donkey.

 It has been removed.

Line 320-330: These sentences have to be move and match with “Conclusion” section or delete. Again, change “phyletic” with “phylogenetic”!

It was moved in the conclusion and changed.

We thank the reveiwer for the important contribution to the improvement of the article. All your concerns have been addressed.

1: In the previous review, I asked a table with polymorphic sites of the mtDNA sequences, according to the reference sequence X97337 and with absolute and relative haplotype frequencies. In your new version I do not found the table that is a mandatory point.

We apologize for this misunderstanding, according to the reviewer request, we added a table (new table 2) including absolute and relative haplotype frequencies.

2: Figure 2: the figure is really too small. I understand that the number refer to the SNPs identified, but they are not the real position on reference sequence X97337, so is impossible compare this figure with literature data. The figure have to be modify.

We thank the reviewer to highlight this point. We modified figure 2 in order to overcome these concerns: SNP numbers refer now to real position on the reference sequence. Plot and text were increased in size.

3: the Authors added sequence of African donkey Somalicus and Nubianus, Chinese ones and other Asiatic Equus kiang, E. hemionus and E. hemionus kulan, this is interesting but do not clarify the history of Italian donkeys. They should align the sequences of this study with those present in GenBank for the same breeds or for other Italian donkey breeds/populations, in order to improve the information about the genetic diversity of the Italian donkey breeds and in Catalonian donkey. I think this is mandatory and one of the goals of the paper.

We added the ClustalW alignment Supplementary materials with other Italian donkey, Tab B see line 133 and Fig B Italian alignment line 194

4: The reference sequence for the median-joining network analysis should be X97337, not the crossbred donkey. Please make again the analyses with the correct reference.

It was done using as reference X97337 which share the same haplotype with Ragusano donkey, now it has been clearly identify by x, see line 208. The reference sequence X97337 is indicated as x.

5: please add the GenBank accession numbers for your sequences.

We added in Supplementary materials the Submission ID # 2466755 Tab C see line 149

6: The sequences used for the NJ have to be listed in “Materials and Methods” section or added as Supplementary materials.

We added sequences in Supplementary materials Tab C see line 140.

7: Please specify in “Materials and Methods” section the software used for the analyses in supplementary figure A.

Supplementary material figure A and B were performed by using itol.embl.de/tree [Ci-tation: Letunic and Bork (2021) Nucleic Acids Res doi: 10.1093/nar/gkab301].

Other comments:

Line 179-182: are you sure that your typical haplotypes are not present in other Italian donkey breeds? For this reason, I suggest the alignment with other sequences from Italian donkey from GenBank.

Accordingly, we removed the world private and as already done, see major points, we perform alignment with GenBank

Line 189: please change “3.2.3. Origin and phyletic relationships” in “3.2.3. Origin and phylogenetic relationships”

done

Line 190: please change “The phyletic relationships…” in “The phylogenetic relationships…..”

done

Line 255: “….. iii. the analysis does not distinguish among breeds”.

It has been removed

We agree with the REV in biological terms of free ranging animals, but here we are in the same conditions of dog pure breed were official center perform let say ‘inbreeding’ between a given number of subject ascribable to a given breed; furthermore Pantesco is completely isolated from any other breed consequently as reported by Bannasch DL, Bannasch MJ, Ryun JR, Famula TR, Pedersen NC (2005) Y chromosome haplotype analysis in purebred dogs. Mamm Genome 16(4):273-80. doi: 10.1007/s00335-004-2435-8: “Breed-specific haplotypes were identified for 26 of the 50 breeds, and haplotype sharing between some breeds indicated a common history, with a significant genetic variation across breeds and with geographic origin of the breeds, particularly among breeds of African origin.”

Sorry, but donkey bred is quite different from dog bred. Dog pure breeds are mainly selected using inbreeding matings with a high level of consanguinity and the breed differentiation is high, as reported in many papers using different markers (for example Sechi et al., 2016; https://doi.org/10.1080/1828051X.2016.1248867; Yang et al., 2019 https://doi.org/10.3389/fgene.2019.01174; Plassais et al., 2019 https://doi.org/10.1038/s41467-019-09373-w). The selective pressure operated by breeders is not so high in donkeys as in horses, so the haplotypes of mtDNA are shared and is impossible identify the breed by the haplotype, as well reported by Stanisic et al 2020 for the Banat donkey (Stanisic et al., 2020; https://doi.org/10.7717/peerj.8598).

In my opinion, the sentence have to be removed.

Line 283: please change “phyletic” with “phylogenetic”.

done

Line 286-391: Please clarify because the circular NJ in figure A is unclear, and I do not found evidence of a Chinese line and two African lines, Nubian and Somali, in your data.

It has been removed.

Line 317-318: This sentence is wrong and have to be removed. It is well known that, over time, PT and RG have repeatedly crossbred each other and with the MF or with the Catalan donkey.

 It has been removed.

Line 320-330: These sentences have to be move and match with “Conclusion” section or delete. Again, change “phyletic” with “phylogenetic”!

It was moved in the conclusion and changed.

We thank the reveiwer for the important contribution to the improvement of the article. All your concerns have been addressed.

1: In the previous review, I asked a table with polymorphic sites of the mtDNA sequences, according to the reference sequence X97337 and with absolute and relative haplotype frequencies. In your new version I do not found the table that is a mandatory point.

We apologize for this misunderstanding, according to the reviewer request, we added a table (new table 2) including absolute and relative haplotype frequencies.

2: Figure 2: the figure is really too small. I understand that the number refer to the SNPs identified, but they are not the real position on reference sequence X97337, so is impossible compare this figure with literature data. The figure have to be modify.

We thank the reviewer to highlight this point. We modified figure 2 in order to overcome these concerns: SNP numbers refer now to real position on the reference sequence. Plot and text were increased in size.

3: the Authors added sequence of African donkey Somalicus and Nubianus, Chinese ones and other Asiatic Equus kiang, E. hemionus and E. hemionus kulan, this is interesting but do not clarify the history of Italian donkeys. They should align the sequences of this study with those present in GenBank for the same breeds or for other Italian donkey breeds/populations, in order to improve the information about the genetic diversity of the Italian donkey breeds and in Catalonian donkey. I think this is mandatory and one of the goals of the paper.

We added the ClustalW alignment Supplementary materials with other Italian donkey, Tab B see line 133 and Fig B Italian alignment line 194

4: The reference sequence for the median-joining network analysis should be X97337, not the crossbred donkey. Please make again the analyses with the correct reference.

It was done using as reference X97337 which share the same haplotype with Ragusano donkey, now it has been clearly identify by x, see line 208. The reference sequence X97337 is indicated as x.

5: please add the GenBank accession numbers for your sequences.

We added in Supplementary materials the Submission ID # 2466755 Tab C see line 149

6: The sequences used for the NJ have to be listed in “Materials and Methods” section or added as Supplementary materials.

We added sequences in Supplementary materials Tab C see line 140.

7: Please specify in “Materials and Methods” section the software used for the analyses in supplementary figure A.

Supplementary material figure A and B were performed by using itol.embl.de/tree [Ci-tation: Letunic and Bork (2021) Nucleic Acids Res doi: 10.1093/nar/gkab301].

Other comments:

Line 179-182: are you sure that your typical haplotypes are not present in other Italian donkey breeds? For this reason, I suggest the alignment with other sequences from Italian donkey from GenBank.

Accordingly, we removed the world private and as already done, see major points, we perform alignment with GenBank

Line 189: please change “3.2.3. Origin and phyletic relationships” in “3.2.3. Origin and phylogenetic relationships”

done

Line 190: please change “The phyletic relationships…” in “The phylogenetic relationships…..”

done

Line 255: “….. iii. the analysis does not distinguish among breeds”.

It has been removed

We agree with the REV in biological terms of free ranging animals, but here we are in the same conditions of dog pure breed were official center perform let say ‘inbreeding’ between a given number of subject ascribable to a given breed; furthermore Pantesco is completely isolated from any other breed consequently as reported by Bannasch DL, Bannasch MJ, Ryun JR, Famula TR, Pedersen NC (2005) Y chromosome haplotype analysis in purebred dogs. Mamm Genome 16(4):273-80. doi: 10.1007/s00335-004-2435-8: “Breed-specific haplotypes were identified for 26 of the 50 breeds, and haplotype sharing between some breeds indicated a common history, with a significant genetic variation across breeds and with geographic origin of the breeds, particularly among breeds of African origin.”

Sorry, but donkey bred is quite different from dog bred. Dog pure breeds are mainly selected using inbreeding matings with a high level of consanguinity and the breed differentiation is high, as reported in many papers using different markers (for example Sechi et al., 2016; https://doi.org/10.1080/1828051X.2016.1248867; Yang et al., 2019 https://doi.org/10.3389/fgene.2019.01174; Plassais et al., 2019 https://doi.org/10.1038/s41467-019-09373-w). The selective pressure operated by breeders is not so high in donkeys as in horses, so the haplotypes of mtDNA are shared and is impossible identify the breed by the haplotype, as well reported by Stanisic et al 2020 for the Banat donkey (Stanisic et al., 2020; https://doi.org/10.7717/peerj.8598).

In my opinion, the sentence have to be removed.

Line 283: please change “phyletic” with “phylogenetic”.

done

Line 286-391: Please clarify because the circular NJ in figure A is unclear, and I do not found evidence of a Chinese line and two African lines, Nubian and Somali, in your data.

It has been removed.

Line 317-318: This sentence is wrong and have to be removed. It is well known that, over time, PT and RG have repeatedly crossbred each other and with the MF or with the Catalan donkey.

 It has been removed.

Line 320-330: These sentences have to be move and match with “Conclusion” section or delete. Again, change “phyletic” with “phylogenetic”!

It was moved in the conclusion and changed.

We thank the reveiwer for the important contribution to the improvement of the article. All your concerns have been addressed.

1: In the previous review, I asked a table with polymorphic sites of the mtDNA sequences, according to the reference sequence X97337 and with absolute and relative haplotype frequencies. In your new version I do not found the table that is a mandatory point.

We apologize for this misunderstanding, according to the reviewer request, we added a table (new table 2) including absolute and relative haplotype frequencies.

2: Figure 2: the figure is really too small. I understand that the number refer to the SNPs identified, but they are not the real position on reference sequence X97337, so is impossible compare this figure with literature data. The figure have to be modify.

We thank the reviewer to highlight this point. We modified figure 2 in order to overcome these concerns: SNP numbers refer now to real position on the reference sequence. Plot and text were increased in size.

3: the Authors added sequence of African donkey Somalicus and Nubianus, Chinese ones and other Asiatic Equus kiang, E. hemionus and E. hemionus kulan, this is interesting but do not clarify the history of Italian donkeys. They should align the sequences of this study with those present in GenBank for the same breeds or for other Italian donkey breeds/populations, in order to improve the information about the genetic diversity of the Italian donkey breeds and in Catalonian donkey. I think this is mandatory and one of the goals of the paper.

We added the ClustalW alignment Supplementary materials with other Italian donkey, Tab B see line 133 and Fig B Italian alignment line 194

4: The reference sequence for the median-joining network analysis should be X97337, not the crossbred donkey. Please make again the analyses with the correct reference.

It was done using as reference X97337 which share the same haplotype with Ragusano donkey, now it has been clearly identify by x, see line 208. The reference sequence X97337 is indicated as x.

5: please add the GenBank accession numbers for your sequences.

We added in Supplementary materials the Submission ID # 2466755 Tab C see line 149

6: The sequences used for the NJ have to be listed in “Materials and Methods” section or added as Supplementary materials.

We added sequences in Supplementary materials Tab C see line 140.

7: Please specify in “Materials and Methods” section the software used for the analyses in supplementary figure A.

Supplementary material figure A and B were performed by using itol.embl.de/tree [Ci-tation: Letunic and Bork (2021) Nucleic Acids Res doi: 10.1093/nar/gkab301].

Other comments:

Line 179-182: are you sure that your typical haplotypes are not present in other Italian donkey breeds? For this reason, I suggest the alignment with other sequences from Italian donkey from GenBank.

Accordingly, we removed the world private and as already done, see major points, we perform alignment with GenBank

Line 189: please change “3.2.3. Origin and phyletic relationships” in “3.2.3. Origin and phylogenetic relationships”

done

Line 190: please change “The phyletic relationships…” in “The phylogenetic relationships…..”

done

Line 255: “….. iii. the analysis does not distinguish among breeds”.

It has been removed

We agree with the REV in biological terms of free ranging animals, but here we are in the same conditions of dog pure breed were official center perform let say ‘inbreeding’ between a given number of subject ascribable to a given breed; furthermore Pantesco is completely isolated from any other breed consequently as reported by Bannasch DL, Bannasch MJ, Ryun JR, Famula TR, Pedersen NC (2005) Y chromosome haplotype analysis in purebred dogs. Mamm Genome 16(4):273-80. doi: 10.1007/s00335-004-2435-8: “Breed-specific haplotypes were identified for 26 of the 50 breeds, and haplotype sharing between some breeds indicated a common history, with a significant genetic variation across breeds and with geographic origin of the breeds, particularly among breeds of African origin.”

Sorry, but donkey bred is quite different from dog bred. Dog pure breeds are mainly selected using inbreeding matings with a high level of consanguinity and the breed differentiation is high, as reported in many papers using different markers (for example Sechi et al., 2016; https://doi.org/10.1080/1828051X.2016.1248867; Yang et al., 2019 https://doi.org/10.3389/fgene.2019.01174; Plassais et al., 2019 https://doi.org/10.1038/s41467-019-09373-w). The selective pressure operated by breeders is not so high in donkeys as in horses, so the haplotypes of mtDNA are shared and is impossible identify the breed by the haplotype, as well reported by Stanisic et al 2020 for the Banat donkey (Stanisic et al., 2020; https://doi.org/10.7717/peerj.8598).

In my opinion, the sentence have to be removed.

Line 283: please change “phyletic” with “phylogenetic”.

done

Line 286-391: Please clarify because the circular NJ in figure A is unclear, and I do not found evidence of a Chinese line and two African lines, Nubian and Somali, in your data.

It has been removed.

Line 317-318: This sentence is wrong and have to be removed. It is well known that, over time, PT and RG have repeatedly crossbred each other and with the MF or with the Catalan donkey.

 It has been removed.

Line 320-330: These sentences have to be move and match with “Conclusion” section or delete. Again, change “phyletic” with “phylogenetic”!

It was moved in the conclusion and changed.

We thank the reveiwer for the important contribution to the improvement of the article. All your concerns have been addressed.

1: In the previous review, I asked a table with polymorphic sites of the mtDNA sequences, according to the reference sequence X97337 and with absolute and relative haplotype frequencies. In your new version I do not found the table that is a mandatory point.

We apologize for this misunderstanding, according to the reviewer request, we added a table (new table 2) including absolute and relative haplotype frequencies.

2: Figure 2: the figure is really too small. I understand that the number refer to the SNPs identified, but they are not the real position on reference sequence X97337, so is impossible compare this figure with literature data. The figure have to be modify.

We thank the reviewer to highlight this point. We modified figure 2 in order to overcome these concerns: SNP numbers refer now to real position on the reference sequence. Plot and text were increased in size.

3: the Authors added sequence of African donkey Somalicus and Nubianus, Chinese ones and other Asiatic Equus kiang, E. hemionus and E. hemionus kulan, this is interesting but do not clarify the history of Italian donkeys. They should align the sequences of this study with those present in GenBank for the same breeds or for other Italian donkey breeds/populations, in order to improve the information about the genetic diversity of the Italian donkey breeds and in Catalonian donkey. I think this is mandatory and one of the goals of the paper.

We added the ClustalW alignment Supplementary materials with other Italian donkey, Tab B see line 133 and Fig B Italian alignment line 194

4: The reference sequence for the median-joining network analysis should be X97337, not the crossbred donkey. Please make again the analyses with the correct reference.

It was done using as reference X97337 which share the same haplotype with Ragusano donkey, now it has been clearly identify by x, see line 208. The reference sequence X97337 is indicated as x.

5: please add the GenBank accession numbers for your sequences.

We added in Supplementary materials the Submission ID # 2466755 Tab C see line 149

6: The sequences used for the NJ have to be listed in “Materials and Methods” section or added as Supplementary materials.

We added sequences in Supplementary materials Tab C see line 140.

7: Please specify in “Materials and Methods” section the software used for the analyses in supplementary figure A.

Supplementary material figure A and B were performed by using itol.embl.de/tree [Ci-tation: Letunic and Bork (2021) Nucleic Acids Res doi: 10.1093/nar/gkab301].

Other comments:

Line 179-182: are you sure that your typical haplotypes are not present in other Italian donkey breeds? For this reason, I suggest the alignment with other sequences from Italian donkey from GenBank.

Accordingly, we removed the world private and as already done, see major points, we perform alignment with GenBank

Line 189: please change “3.2.3. Origin and phyletic relationships” in “3.2.3. Origin and phylogenetic relationships”

done

Line 190: please change “The phyletic relationships…” in “The phylogenetic relationships…..”

done

Line 255: “….. iii. the analysis does not distinguish among breeds”.

It has been removed

We agree with the REV in biological terms of free ranging animals, but here we are in the same conditions of dog pure breed were official center perform let say ‘inbreeding’ between a given number of subject ascribable to a given breed; furthermore Pantesco is completely isolated from any other breed consequently as reported by Bannasch DL, Bannasch MJ, Ryun JR, Famula TR, Pedersen NC (2005) Y chromosome haplotype analysis in purebred dogs. Mamm Genome 16(4):273-80. doi: 10.1007/s00335-004-2435-8: “Breed-specific haplotypes were identified for 26 of the 50 breeds, and haplotype sharing between some breeds indicated a common history, with a significant genetic variation across breeds and with geographic origin of the breeds, particularly among breeds of African origin.”

Sorry, but donkey bred is quite different from dog bred. Dog pure breeds are mainly selected using inbreeding matings with a high level of consanguinity and the breed differentiation is high, as reported in many papers using different markers (for example Sechi et al., 2016; https://doi.org/10.1080/1828051X.2016.1248867; Yang et al., 2019 https://doi.org/10.3389/fgene.2019.01174; Plassais et al., 2019 https://doi.org/10.1038/s41467-019-09373-w). The selective pressure operated by breeders is not so high in donkeys as in horses, so the haplotypes of mtDNA are shared and is impossible identify the breed by the haplotype, as well reported by Stanisic et al 2020 for the Banat donkey (Stanisic et al., 2020; https://doi.org/10.7717/peerj.8598).

In my opinion, the sentence have to be removed.

Line 283: please change “phyletic” with “phylogenetic”.

done

Line 286-391: Please clarify because the circular NJ in figure A is unclear, and I do not found evidence of a Chinese line and two African lines, Nubian and Somali, in your data.

It has been removed.

Line 317-318: This sentence is wrong and have to be removed. It is well known that, over time, PT and RG have repeatedly crossbred each other and with the MF or with the Catalan donkey.

 It has been removed.

Line 320-330: These sentences have to be move and match with “Conclusion” section or delete. Again, change “phyletic” with “phylogenetic”!

It was moved in the conclusion and changed.

We thank the reveiwer for the important contribution to the improvement of the article. All your concerns have been addressed.

1: In the previous review, I asked a table with polymorphic sites of the mtDNA sequences, according to the reference sequence X97337 and with absolute and relative haplotype frequencies. In your new version I do not found the table that is a mandatory point.

We apologize for this misunderstanding, according to the reviewer request, we added a table (new table 2) including absolute and relative haplotype frequencies.

2: Figure 2: the figure is really too small. I understand that the number refer to the SNPs identified, but they are not the real position on reference sequence X97337, so is impossible compare this figure with literature data. The figure have to be modify.

We thank the reviewer to highlight this point. We modified figure 2 in order to overcome these concerns: SNP numbers refer now to real position on the reference sequence. Plot and text were increased in size.

3: the Authors added sequence of African donkey Somalicus and Nubianus, Chinese ones and other Asiatic Equus kiang, E. hemionus and E. hemionus kulan, this is interesting but do not clarify the history of Italian donkeys. They should align the sequences of this study with those present in GenBank for the same breeds or for other Italian donkey breeds/populations, in order to improve the information about the genetic diversity of the Italian donkey breeds and in Catalonian donkey. I think this is mandatory and one of the goals of the paper.

We added the ClustalW alignment Supplementary materials with other Italian donkey, Tab B see line 133 and Fig B Italian alignment line 194

4: The reference sequence for the median-joining network analysis should be X97337, not the crossbred donkey. Please make again the analyses with the correct reference.

It was done using as reference X97337 which share the same haplotype with Ragusano donkey, now it has been clearly identify by x, see line 208. The reference sequence X97337 is indicated as x.

5: please add the GenBank accession numbers for your sequences.

We added in Supplementary materials the Submission ID # 2466755 Tab C see line 149

6: The sequences used for the NJ have to be listed in “Materials and Methods” section or added as Supplementary materials.

We added sequences in Supplementary materials Tab C see line 140.

7: Please specify in “Materials and Methods” section the software used for the analyses in supplementary figure A.

Supplementary material figure A and B were performed by using itol.embl.de/tree [Ci-tation: Letunic and Bork (2021) Nucleic Acids Res doi: 10.1093/nar/gkab301].

Other comments:

Line 179-182: are you sure that your typical haplotypes are not present in other Italian donkey breeds? For this reason, I suggest the alignment with other sequences from Italian donkey from GenBank.

Accordingly, we removed the world private and as already done, see major points, we perform alignment with GenBank

Line 189: please change “3.2.3. Origin and phyletic relationships” in “3.2.3. Origin and phylogenetic relationships”

done

Line 190: please change “The phyletic relationships…” in “The phylogenetic relationships…..”

done

Line 255: “….. iii. the analysis does not distinguish among breeds”.

It has been removed

We agree with the REV in biological terms of free ranging animals, but here we are in the same conditions of dog pure breed were official center perform let say ‘inbreeding’ between a given number of subject ascribable to a given breed; furthermore Pantesco is completely isolated from any other breed consequently as reported by Bannasch DL, Bannasch MJ, Ryun JR, Famula TR, Pedersen NC (2005) Y chromosome haplotype analysis in purebred dogs. Mamm Genome 16(4):273-80. doi: 10.1007/s00335-004-2435-8: “Breed-specific haplotypes were identified for 26 of the 50 breeds, and haplotype sharing between some breeds indicated a common history, with a significant genetic variation across breeds and with geographic origin of the breeds, particularly among breeds of African origin.”

Sorry, but donkey bred is quite different from dog bred. Dog pure breeds are mainly selected using inbreeding matings with a high level of consanguinity and the breed differentiation is high, as reported in many papers using different markers (for example Sechi et al., 2016; https://doi.org/10.1080/1828051X.2016.1248867; Yang et al., 2019 https://doi.org/10.3389/fgene.2019.01174; Plassais et al., 2019 https://doi.org/10.1038/s41467-019-09373-w). The selective pressure operated by breeders is not so high in donkeys as in horses, so the haplotypes of mtDNA are shared and is impossible identify the breed by the haplotype, as well reported by Stanisic et al 2020 for the Banat donkey (Stanisic et al., 2020; https://doi.org/10.7717/peerj.8598).

In my opinion, the sentence have to be removed.

Line 283: please change “phyletic” with “phylogenetic”.

done

Line 286-391: Please clarify because the circular NJ in figure A is unclear, and I do not found evidence of a Chinese line and two African lines, Nubian and Somali, in your data.

It has been removed.

Line 317-318: This sentence is wrong and have to be removed. It is well known that, over time, PT and RG have repeatedly crossbred each other and with the MF or with the Catalan donkey.

 It has been removed.

Line 320-330: These sentences have to be move and match with “Conclusion” section or delete. Again, change “phyletic” with “phylogenetic”!

It was moved in the conclusion and changed.

We thank the reveiwer for the important contribution to the improvement of the article. All your concerns have been addressed.

1: In the previous review, I asked a table with polymorphic sites of the mtDNA sequences, according to the reference sequence X97337 and with absolute and relative haplotype frequencies. In your new version I do not found the table that is a mandatory point.

We apologize for this misunderstanding, according to the reviewer request, we added a table (new table 2) including absolute and relative haplotype frequencies.

2: Figure 2: the figure is really too small. I understand that the number refer to the SNPs identified, but they are not the real position on reference sequence X97337, so is impossible compare this figure with literature data. The figure have to be modify.

We thank the reviewer to highlight this point. We modified figure 2 in order to overcome these concerns: SNP numbers refer now to real position on the reference sequence. Plot and text were increased in size.

3: the Authors added sequence of African donkey Somalicus and Nubianus, Chinese ones and other Asiatic Equus kiang, E. hemionus and E. hemionus kulan, this is interesting but do not clarify the history of Italian donkeys. They should align the sequences of this study with those present in GenBank for the same breeds or for other Italian donkey breeds/populations, in order to improve the information about the genetic diversity of the Italian donkey breeds and in Catalonian donkey. I think this is mandatory and one of the goals of the paper.

We added the ClustalW alignment Supplementary materials with other Italian donkey, Tab B see line 133 and Fig B Italian alignment line 194

4: The reference sequence for the median-joining network analysis should be X97337, not the crossbred donkey. Please make again the analyses with the correct reference.

It was done using as reference X97337 which share the same haplotype with Ragusano donkey, now it has been clearly identify by x, see line 208. The reference sequence X97337 is indicated as x.

5: please add the GenBank accession numbers for your sequences.

We added in Supplementary materials the Submission ID # 2466755 Tab C see line 149

6: The sequences used for the NJ have to be listed in “Materials and Methods” section or added as Supplementary materials.

We added sequences in Supplementary materials Tab C see line 140.

7: Please specify in “Materials and Methods” section the software used for the analyses in supplementary figure A.

Supplementary material figure A and B were performed by using itol.embl.de/tree [Ci-tation: Letunic and Bork (2021) Nucleic Acids Res doi: 10.1093/nar/gkab301].

Other comments:

Line 179-182: are you sure that your typical haplotypes are not present in other Italian donkey breeds? For this reason, I suggest the alignment with other sequences from Italian donkey from GenBank.

Accordingly, we removed the world private and as already done, see major points, we perform alignment with GenBank

Line 189: please change “3.2.3. Origin and phyletic relationships” in “3.2.3. Origin and phylogenetic relationships”

done

Line 190: please change “The phyletic relationships…” in “The phylogenetic relationships…..”

done

Line 255: “….. iii. the analysis does not distinguish among breeds”.

It has been removed

We agree with the REV in biological terms of free ranging animals, but here we are in the same conditions of dog pure breed were official center perform let say ‘inbreeding’ between a given number of subject ascribable to a given breed; furthermore Pantesco is completely isolated from any other breed consequently as reported by Bannasch DL, Bannasch MJ, Ryun JR, Famula TR, Pedersen NC (2005) Y chromosome haplotype analysis in purebred dogs. Mamm Genome 16(4):273-80. doi: 10.1007/s00335-004-2435-8: “Breed-specific haplotypes were identified for 26 of the 50 breeds, and haplotype sharing between some breeds indicated a common history, with a significant genetic variation across breeds and with geographic origin of the breeds, particularly among breeds of African origin.”

Sorry, but donkey bred is quite different from dog bred. Dog pure breeds are mainly selected using inbreeding matings with a high level of consanguinity and the breed differentiation is high, as reported in many papers using different markers (for example Sechi et al., 2016; https://doi.org/10.1080/1828051X.2016.1248867; Yang et al., 2019 https://doi.org/10.3389/fgene.2019.01174; Plassais et al., 2019 https://doi.org/10.1038/s41467-019-09373-w). The selective pressure operated by breeders is not so high in donkeys as in horses, so the haplotypes of mtDNA are shared and is impossible identify the breed by the haplotype, as well reported by Stanisic et al 2020 for the Banat donkey (Stanisic et al., 2020; https://doi.org/10.7717/peerj.8598).

In my opinion, the sentence have to be removed.

Line 283: please change “phyletic” with “phylogenetic”.

done

Line 286-391: Please clarify because the circular NJ in figure A is unclear, and I do not found evidence of a Chinese line and two African lines, Nubian and Somali, in your data.

It has been removed.

Line 317-318: This sentence is wrong and have to be removed. It is well known that, over time, PT and RG have repeatedly crossbred each other and with the MF or with the Catalan donkey.

 It has been removed.

Line 320-330: These sentences have to be move and match with “Conclusion” section or delete. Again, change “phyletic” with “phylogenetic”!

It was moved in the conclusion and changed.

We thank the reveiwer for the important contribution to the improvement of the article. All your concerns have been addressed.

1: In the previous review, I asked a table with polymorphic sites of the mtDNA sequences, according to the reference sequence X97337 and with absolute and relative haplotype frequencies. In your new version I do not found the table that is a mandatory point.

We apologize for this misunderstanding, according to the reviewer request, we added a table (new table 2) including absolute and relative haplotype frequencies.

2: Figure 2: the figure is really too small. I understand that the number refer to the SNPs identified, but they are not the real position on reference sequence X97337, so is impossible compare this figure with literature data. The figure have to be modify.

We thank the reviewer to highlight this point. We modified figure 2 in order to overcome these concerns: SNP numbers refer now to real position on the reference sequence. Plot and text were increased in size.

3: the Authors added sequence of African donkey Somalicus and Nubianus, Chinese ones and other Asiatic Equus kiang, E. hemionus and E. hemionus kulan, this is interesting but do not clarify the history of Italian donkeys. They should align the sequences of this study with those present in GenBank for the same breeds or for other Italian donkey breeds/populations, in order to improve the information about the genetic diversity of the Italian donkey breeds and in Catalonian donkey. I think this is mandatory and one of the goals of the paper.

We added the ClustalW alignment Supplementary materials with other Italian donkey, Tab B see line 133 and Fig B Italian alignment line 194

4: The reference sequence for the median-joining network analysis should be X97337, not the crossbred donkey. Please make again the analyses with the correct reference.

It was done using as reference X97337 which share the same haplotype with Ragusano donkey, now it has been clearly identify by x, see line 208. The reference sequence X97337 is indicated as x.

5: please add the GenBank accession numbers for your sequences.

We added in Supplementary materials the Submission ID # 2466755 Tab C see line 149

6: The sequences used for the NJ have to be listed in “Materials and Methods” section or added as Supplementary materials.

We added sequences in Supplementary materials Tab C see line 140.

7: Please specify in “Materials and Methods” section the software used for the analyses in supplementary figure A.

Supplementary material figure A and B were performed by using itol.embl.de/tree [Ci-tation: Letunic and Bork (2021) Nucleic Acids Res doi: 10.1093/nar/gkab301].

Other comments:

Line 179-182: are you sure that your typical haplotypes are not present in other Italian donkey breeds? For this reason, I suggest the alignment with other sequences from Italian donkey from GenBank.

Accordingly, we removed the world private and as already done, see major points, we perform alignment with GenBank

Line 189: please change “3.2.3. Origin and phyletic relationships” in “3.2.3. Origin and phylogenetic relationships”

done

Line 190: please change “The phyletic relationships…” in “The phylogenetic relationships…..”

done

Line 255: “….. iii. the analysis does not distinguish among breeds”.

It has been removed

We agree with the REV in biological terms of free ranging animals, but here we are in the same conditions of dog pure breed were official center perform let say ‘inbreeding’ between a given number of subject ascribable to a given breed; furthermore Pantesco is completely isolated from any other breed consequently as reported by Bannasch DL, Bannasch MJ, Ryun JR, Famula TR, Pedersen NC (2005) Y chromosome haplotype analysis in purebred dogs. Mamm Genome 16(4):273-80. doi: 10.1007/s00335-004-2435-8: “Breed-specific haplotypes were identified for 26 of the 50 breeds, and haplotype sharing between some breeds indicated a common history, with a significant genetic variation across breeds and with geographic origin of the breeds, particularly among breeds of African origin.”

Sorry, but donkey bred is quite different from dog bred. Dog pure breeds are mainly selected using inbreeding matings with a high level of consanguinity and the breed differentiation is high, as reported in many papers using different markers (for example Sechi et al., 2016; https://doi.org/10.1080/1828051X.2016.1248867; Yang et al., 2019 https://doi.org/10.3389/fgene.2019.01174; Plassais et al., 2019 https://doi.org/10.1038/s41467-019-09373-w). The selective pressure operated by breeders is not so high in donkeys as in horses, so the haplotypes of mtDNA are shared and is impossible identify the breed by the haplotype, as well reported by Stanisic et al 2020 for the Banat donkey (Stanisic et al., 2020; https://doi.org/10.7717/peerj.8598).

In my opinion, the sentence have to be removed.

Line 283: please change “phyletic” with “phylogenetic”.

done

Line 286-391: Please clarify because the circular NJ in figure A is unclear, and I do not found evidence of a Chinese line and two African lines, Nubian and Somali, in your data.

It has been removed.

Line 317-318: This sentence is wrong and have to be removed. It is well known that, over time, PT and RG have repeatedly crossbred each other and with the MF or with the Catalan donkey.

 It has been removed.

Line 320-330: These sentences have to be move and match with “Conclusion” section or delete. Again, change “phyletic” with “phylogenetic”!

It was moved in the conclusion and changed.

We thank the reveiwer for the important contribution to the improvement of the article. All your concerns have been addressed.

1: In the previous review, I asked a table with polymorphic sites of the mtDNA sequences, according to the reference sequence X97337 and with absolute and relative haplotype frequencies. In your new version I do not found the table that is a mandatory point.

We apologize for this misunderstanding, according to the reviewer request, we added a table (new table 2) including absolute and relative haplotype frequencies.

2: Figure 2: the figure is really too small. I understand that the number refer to the SNPs identified, but they are not the real position on reference sequence X97337, so is impossible compare this figure with literature data. The figure have to be modify.

We thank the reviewer to highlight this point. We modified figure 2 in order to overcome these concerns: SNP numbers refer now to real position on the reference sequence. Plot and text were increased in size.

3: the Authors added sequence of African donkey Somalicus and Nubianus, Chinese ones and other Asiatic Equus kiang, E. hemionus and E. hemionus kulan, this is interesting but do not clarify the history of Italian donkeys. They should align the sequences of this study with those present in GenBank for the same breeds or for other Italian donkey breeds/populations, in order to improve the information about the genetic diversity of the Italian donkey breeds and in Catalonian donkey. I think this is mandatory and one of the goals of the paper.

We added the ClustalW alignment Supplementary materials with other Italian donkey, Tab B see line 133 and Fig B Italian alignment line 194

4: The reference sequence for the median-joining network analysis should be X97337, not the crossbred donkey. Please make again the analyses with the correct reference.

It was done using as reference X97337 which share the same haplotype with Ragusano donkey, now it has been clearly identify by x, see line 208. The reference sequence X97337 is indicated as x.

5: please add the GenBank accession numbers for your sequences.

We added in Supplementary materials the Submission ID # 2466755 Tab C see line 149

6: The sequences used for the NJ have to be listed in “Materials and Methods” section or added as Supplementary materials.

We added sequences in Supplementary materials Tab C see line 140.

7: Please specify in “Materials and Methods” section the software used for the analyses in supplementary figure A.

Supplementary material figure A and B were performed by using itol.embl.de/tree [Ci-tation: Letunic and Bork (2021) Nucleic Acids Res doi: 10.1093/nar/gkab301].

Other comments:

Line 179-182: are you sure that your typical haplotypes are not present in other Italian donkey breeds? For this reason, I suggest the alignment with other sequences from Italian donkey from GenBank.

Accordingly, we removed the world private and as already done, see major points, we perform alignment with GenBank

Line 189: please change “3.2.3. Origin and phyletic relationships” in “3.2.3. Origin and phylogenetic relationships”

done

Line 190: please change “The phyletic relationships…” in “The phylogenetic relationships…..”

done

Line 255: “….. iii. the analysis does not distinguish among breeds”.

It has been removed

We agree with the REV in biological terms of free ranging animals, but here we are in the same conditions of dog pure breed were official center perform let say ‘inbreeding’ between a given number of subject ascribable to a given breed; furthermore Pantesco is completely isolated from any other breed consequently as reported by Bannasch DL, Bannasch MJ, Ryun JR, Famula TR, Pedersen NC (2005) Y chromosome haplotype analysis in purebred dogs. Mamm Genome 16(4):273-80. doi: 10.1007/s00335-004-2435-8: “Breed-specific haplotypes were identified for 26 of the 50 breeds, and haplotype sharing between some breeds indicated a common history, with a significant genetic variation across breeds and with geographic origin of the breeds, particularly among breeds of African origin.”

Sorry, but donkey bred is quite different from dog bred. Dog pure breeds are mainly selected using inbreeding matings with a high level of consanguinity and the breed differentiation is high, as reported in many papers using different markers (for example Sechi et al., 2016; https://doi.org/10.1080/1828051X.2016.1248867; Yang et al., 2019 https://doi.org/10.3389/fgene.2019.01174; Plassais et al., 2019 https://doi.org/10.1038/s41467-019-09373-w). The selective pressure operated by breeders is not so high in donkeys as in horses, so the haplotypes of mtDNA are shared and is impossible identify the breed by the haplotype, as well reported by Stanisic et al 2020 for the Banat donkey (Stanisic et al., 2020; https://doi.org/10.7717/peerj.8598).

In my opinion, the sentence have to be removed.

Line 283: please change “phyletic” with “phylogenetic”.

done

Line 286-391: Please clarify because the circular NJ in figure A is unclear, and I do not found evidence of a Chinese line and two African lines, Nubian and Somali, in your data.

It has been removed.

Line 317-318: This sentence is wrong and have to be removed. It is well known that, over time, PT and RG have repeatedly crossbred each other and with the MF or with the Catalan donkey.

 It has been removed.

Line 320-330: These sentences have to be move and match with “Conclusion” section or delete. Again, change “phyletic” with “phylogenetic”!

It was moved in the conclusion and changed.

We thank the reveiwer for the important contribution to the improvement of the article. All your concerns have been addressed.

1: In the previous review, I asked a table with polymorphic sites of the mtDNA sequences, according to the reference sequence X97337 and with absolute and relative haplotype frequencies. In your new version I do not found the table that is a mandatory point.

We apologize for this misunderstanding, according to the reviewer request, we added a table (new table 2) including absolute and relative haplotype frequencies.

2: Figure 2: the figure is really too small. I understand that the number refer to the SNPs identified, but they are not the real position on reference sequence X97337, so is impossible compare this figure with literature data. The figure have to be modify.

We thank the reviewer to highlight this point. We modified figure 2 in order to overcome these concerns: SNP numbers refer now to real position on the reference sequence. Plot and text were increased in size.

3: the Authors added sequence of African donkey Somalicus and Nubianus, Chinese ones and other Asiatic Equus kiang, E. hemionus and E. hemionus kulan, this is interesting but do not clarify the history of Italian donkeys. They should align the sequences of this study with those present in GenBank for the same breeds or for other Italian donkey breeds/populations, in order to improve the information about the genetic diversity of the Italian donkey breeds and in Catalonian donkey. I think this is mandatory and one of the goals of the paper.

We added the ClustalW alignment Supplementary materials with other Italian donkey, Tab B see line 133 and Fig B Italian alignment line 194

4: The reference sequence for the median-joining network analysis should be X97337, not the crossbred donkey. Please make again the analyses with the correct reference.

It was done using as reference X97337 which share the same haplotype with Ragusano donkey, now it has been clearly identify by x, see line 208. The reference sequence X97337 is indicated as x.

5: please add the GenBank accession numbers for your sequences.

We added in Supplementary materials the Submission ID # 2466755 Tab C see line 149

6: The sequences used for the NJ have to be listed in “Materials and Methods” section or added as Supplementary materials.

We added sequences in Supplementary materials Tab C see line 140.

7: Please specify in “Materials and Methods” section the software used for the analyses in supplementary figure A.

Supplementary material figure A and B were performed by using itol.embl.de/tree [Ci-tation: Letunic and Bork (2021) Nucleic Acids Res doi: 10.1093/nar/gkab301].

Other comments:

Line 179-182: are you sure that your typical haplotypes are not present in other Italian donkey breeds? For this reason, I suggest the alignment with other sequences from Italian donkey from GenBank.

Accordingly, we removed the world private and as already done, see major points, we perform alignment with GenBank

Line 189: please change “3.2.3. Origin and phyletic relationships” in “3.2.3. Origin and phylogenetic relationships”

done

Line 190: please change “The phyletic relationships…” in “The phylogenetic relationships…..”

done

Line 255: “….. iii. the analysis does not distinguish among breeds”.

It has been removed

We agree with the REV in biological terms of free ranging animals, but here we are in the same conditions of dog pure breed were official center perform let say ‘inbreeding’ between a given number of subject ascribable to a given breed; furthermore Pantesco is completely isolated from any other breed consequently as reported by Bannasch DL, Bannasch MJ, Ryun JR, Famula TR, Pedersen NC (2005) Y chromosome haplotype analysis in purebred dogs. Mamm Genome 16(4):273-80. doi: 10.1007/s00335-004-2435-8: “Breed-specific haplotypes were identified for 26 of the 50 breeds, and haplotype sharing between some breeds indicated a common history, with a significant genetic variation across breeds and with geographic origin of the breeds, particularly among breeds of African origin.”

Sorry, but donkey bred is quite different from dog bred. Dog pure breeds are mainly selected using inbreeding matings with a high level of consanguinity and the breed differentiation is high, as reported in many papers using different markers (for example Sechi et al., 2016; https://doi.org/10.1080/1828051X.2016.1248867; Yang et al., 2019 https://doi.org/10.3389/fgene.2019.01174; Plassais et al., 2019 https://doi.org/10.1038/s41467-019-09373-w). The selective pressure operated by breeders is not so high in donkeys as in horses, so the haplotypes of mtDNA are shared and is impossible identify the breed by the haplotype, as well reported by Stanisic et al 2020 for the Banat donkey (Stanisic et al., 2020; https://doi.org/10.7717/peerj.8598).

In my opinion, the sentence have to be removed.

Line 283: please change “phyletic” with “phylogenetic”.

done

Line 286-391: Please clarify because the circular NJ in figure A is unclear, and I do not found evidence of a Chinese line and two African lines, Nubian and Somali, in your data.

It has been removed.

Line 317-318: This sentence is wrong and have to be removed. It is well known that, over time, PT and RG have repeatedly crossbred each other and with the MF or with the Catalan donkey.

 It has been removed.

Line 320-330: These sentences have to be move and match with “Conclusion” section or delete. Again, change “phyletic” with “phylogenetic”!

It was moved in the conclusion and changed.

We thank the reveiwer for the important contribution to the improvement of the article. All your concerns have been addressed.

1: In the previous review, I asked a table with polymorphic sites of the mtDNA sequences, according to the reference sequence X97337 and with absolute and relative haplotype frequencies. In your new version I do not found the table that is a mandatory point.

We apologize for this misunderstanding, according to the reviewer request, we added a table (new table 2) including absolute and relative haplotype frequencies.

2: Figure 2: the figure is really too small. I understand that the number refer to the SNPs identified, but they are not the real position on reference sequence X97337, so is impossible compare this figure with literature data. The figure have to be modify.

We thank the reviewer to highlight this point. We modified figure 2 in order to overcome these concerns: SNP numbers refer now to real position on the reference sequence. Plot and text were increased in size.

3: the Authors added sequence of African donkey Somalicus and Nubianus, Chinese ones and other Asiatic Equus kiang, E. hemionus and E. hemionus kulan, this is interesting but do not clarify the history of Italian donkeys. They should align the sequences of this study with those present in GenBank for the same breeds or for other Italian donkey breeds/populations, in order to improve the information about the genetic diversity of the Italian donkey breeds and in Catalonian donkey. I think this is mandatory and one of the goals of the paper.

We added the ClustalW alignment Supplementary materials with other Italian donkey, Tab B see line 133 and Fig B Italian alignment line 194

4: The reference sequence for the median-joining network analysis should be X97337, not the crossbred donkey. Please make again the analyses with the correct reference.

It was done using as reference X97337 which share the same haplotype with Ragusano donkey, now it has been clearly identify by x, see line 208. The reference sequence X97337 is indicated as x.

5: please add the GenBank accession numbers for your sequences.

We added in Supplementary materials the Submission ID # 2466755 Tab C see line 149

6: The sequences used for the NJ have to be listed in “Materials and Methods” section or added as Supplementary materials.

We added sequences in Supplementary materials Tab C see line 140.

7: Please specify in “Materials and Methods” section the software used for the analyses in supplementary figure A.

Supplementary material figure A and B were performed by using itol.embl.de/tree [Ci-tation: Letunic and Bork (2021) Nucleic Acids Res doi: 10.1093/nar/gkab301].

Other comments:

Line 179-182: are you sure that your typical haplotypes are not present in other Italian donkey breeds? For this reason, I suggest the alignment with other sequences from Italian donkey from GenBank.

Accordingly, we removed the world private and as already done, see major points, we perform alignment with GenBank

Line 189: please change “3.2.3. Origin and phyletic relationships” in “3.2.3. Origin and phylogenetic relationships”

done

Line 190: please change “The phyletic relationships…” in “The phylogenetic relationships…..”

done

Line 255: “….. iii. the analysis does not distinguish among breeds”.

It has been removed

We agree with the REV in biological terms of free ranging animals, but here we are in the same conditions of dog pure breed were official center perform let say ‘inbreeding’ between a given number of subject ascribable to a given breed; furthermore Pantesco is completely isolated from any other breed consequently as reported by Bannasch DL, Bannasch MJ, Ryun JR, Famula TR, Pedersen NC (2005) Y chromosome haplotype analysis in purebred dogs. Mamm Genome 16(4):273-80. doi: 10.1007/s00335-004-2435-8: “Breed-specific haplotypes were identified for 26 of the 50 breeds, and haplotype sharing between some breeds indicated a common history, with a significant genetic variation across breeds and with geographic origin of the breeds, particularly among breeds of African origin.”

Sorry, but donkey bred is quite different from dog bred. Dog pure breeds are mainly selected using inbreeding matings with a high level of consanguinity and the breed differentiation is high, as reported in many papers using different markers (for example Sechi et al., 2016; https://doi.org/10.1080/1828051X.2016.1248867; Yang et al., 2019 https://doi.org/10.3389/fgene.2019.01174; Plassais et al., 2019 https://doi.org/10.1038/s41467-019-09373-w). The selective pressure operated by breeders is not so high in donkeys as in horses, so the haplotypes of mtDNA are shared and is impossible identify the breed by the haplotype, as well reported by Stanisic et al 2020 for the Banat donkey (Stanisic et al., 2020; https://doi.org/10.7717/peerj.8598).

In my opinion, the sentence have to be removed.

Line 283: please change “phyletic” with “phylogenetic”.

done

Line 286-391: Please clarify because the circular NJ in figure A is unclear, and I do not found evidence of a Chinese line and two African lines, Nubian and Somali, in your data.

It has been removed.

Line 317-318: This sentence is wrong and have to be removed. It is well known that, over time, PT and RG have repeatedly crossbred each other and with the MF or with the Catalan donkey.

 It has been removed.

Line 320-330: These sentences have to be move and match with “Conclusion” section or delete. Again, change “phyletic” with “phylogenetic”!

It was moved in the conclusion and changed.

We thank the reveiwer for the important contribution to the improvement of the article. All your concerns have been addressed.

1: In the previous review, I asked a table with polymorphic sites of the mtDNA sequences, according to the reference sequence X97337 and with absolute and relative haplotype frequencies. In your new version I do not found the table that is a mandatory point.

We apologize for this misunderstanding, according to the reviewer request, we added a table (new table 2) including absolute and relative haplotype frequencies.

2: Figure 2: the figure is really too small. I understand that the number refer to the SNPs identified, but they are not the real position on reference sequence X97337, so is impossible compare this figure with literature data. The figure have to be modify.

We thank the reviewer to highlight this point. We modified figure 2 in order to overcome these concerns: SNP numbers refer now to real position on the reference sequence. Plot and text were increased in size.

3: the Authors added sequence of African donkey Somalicus and Nubianus, Chinese ones and other Asiatic Equus kiang, E. hemionus and E. hemionus kulan, this is interesting but do not clarify the history of Italian donkeys. They should align the sequences of this study with those present in GenBank for the same breeds or for other Italian donkey breeds/populations, in order to improve the information about the genetic diversity of the Italian donkey breeds and in Catalonian donkey. I think this is mandatory and one of the goals of the paper.

We added the ClustalW alignment Supplementary materials with other Italian donkey, Tab B see line 133 and Fig B Italian alignment line 194

4: The reference sequence for the median-joining network analysis should be X97337, not the crossbred donkey. Please make again the analyses with the correct reference.

It was done using as reference X97337 which share the same haplotype with Ragusano donkey, now it has been clearly identify by x, see line 208. The reference sequence X97337 is indicated as x.

5: please add the GenBank accession numbers for your sequences.

We added in Supplementary materials the Submission ID # 2466755 Tab C see line 149

6: The sequences used for the NJ have to be listed in “Materials and Methods” section or added as Supplementary materials.

We added sequences in Supplementary materials Tab C see line 140.

7: Please specify in “Materials and Methods” section the software used for the analyses in supplementary figure A.

Supplementary material figure A and B were performed by using itol.embl.de/tree [Ci-tation: Letunic and Bork (2021) Nucleic Acids Res doi: 10.1093/nar/gkab301].

Other comments:

Line 179-182: are you sure that your typical haplotypes are not present in other Italian donkey breeds? For this reason, I suggest the alignment with other sequences from Italian donkey from GenBank.

Accordingly, we removed the world private and as already done, see major points, we perform alignment with GenBank

Line 189: please change “3.2.3. Origin and phyletic relationships” in “3.2.3. Origin and phylogenetic relationships”

done

Line 190: please change “The phyletic relationships…” in “The phylogenetic relationships…..”

done

Line 255: “….. iii. the analysis does not distinguish among breeds”.

It has been removed

We agree with the REV in biological terms of free ranging animals, but here we are in the same conditions of dog pure breed were official center perform let say ‘inbreeding’ between a given number of subject ascribable to a given breed; furthermore Pantesco is completely isolated from any other breed consequently as reported by Bannasch DL, Bannasch MJ, Ryun JR, Famula TR, Pedersen NC (2005) Y chromosome haplotype analysis in purebred dogs. Mamm Genome 16(4):273-80. doi: 10.1007/s00335-004-2435-8: “Breed-specific haplotypes were identified for 26 of the 50 breeds, and haplotype sharing between some breeds indicated a common history, with a significant genetic variation across breeds and with geographic origin of the breeds, particularly among breeds of African origin.”

Sorry, but donkey bred is quite different from dog bred. Dog pure breeds are mainly selected using inbreeding matings with a high level of consanguinity and the breed differentiation is high, as reported in many papers using different markers (for example Sechi et al., 2016; https://doi.org/10.1080/1828051X.2016.1248867; Yang et al., 2019 https://doi.org/10.3389/fgene.2019.01174; Plassais et al., 2019 https://doi.org/10.1038/s41467-019-09373-w). The selective pressure operated by breeders is not so high in donkeys as in horses, so the haplotypes of mtDNA are shared and is impossible identify the breed by the haplotype, as well reported by Stanisic et al 2020 for the Banat donkey (Stanisic et al., 2020; https://doi.org/10.7717/peerj.8598).

In my opinion, the sentence have to be removed.

Line 283: please change “phyletic” with “phylogenetic”.

done

Line 286-391: Please clarify because the circular NJ in figure A is unclear, and I do not found evidence of a Chinese line and two African lines, Nubian and Somali, in your data.

It has been removed.

Line 317-318: This sentence is wrong and have to be removed. It is well known that, over time, PT and RG have repeatedly crossbred each other and with the MF or with the Catalan donkey.

 It has been removed.

Line 320-330: These sentences have to be move and match with “Conclusion” section or delete. Again, change “phyletic” with “phylogenetic”!

It was moved in the conclusion and changed.

We thank the reveiwer for the important contribution to the improvement of the article. All your concerns have been addressed.

1: In the previous review, I asked a table with polymorphic sites of the mtDNA sequences, according to the reference sequence X97337 and with absolute and relative haplotype frequencies. In your new version I do not found the table that is a mandatory point.

We apologize for this misunderstanding, according to the reviewer request, we added a table (new table 2) including absolute and relative haplotype frequencies.

2: Figure 2: the figure is really too small. I understand that the number refer to the SNPs identified, but they are not the real position on reference sequence X97337, so is impossible compare this figure with literature data. The figure have to be modify.

We thank the reviewer to highlight this point. We modified figure 2 in order to overcome these concerns: SNP numbers refer now to real position on the reference sequence. Plot and text were increased in size.

3: the Authors added sequence of African donkey Somalicus and Nubianus, Chinese ones and other Asiatic Equus kiang, E. hemionus and E. hemionus kulan, this is interesting but do not clarify the history of Italian donkeys. They should align the sequences of this study with those present in GenBank for the same breeds or for other Italian donkey breeds/populations, in order to improve the information about the genetic diversity of the Italian donkey breeds and in Catalonian donkey. I think this is mandatory and one of the goals of the paper.

We added the ClustalW alignment Supplementary materials with other Italian donkey, Tab B see line 133 and Fig B Italian alignment line 194

4: The reference sequence for the median-joining network analysis should be X97337, not the crossbred donkey. Please make again the analyses with the correct reference.

It was done using as reference X97337 which share the same haplotype with Ragusano donkey, now it has been clearly identify by x, see line 208. The reference sequence X97337 is indicated as x.

5: please add the GenBank accession numbers for your sequences.

We added in Supplementary materials the Submission ID # 2466755 Tab C see line 149

6: The sequences used for the NJ have to be listed in “Materials and Methods” section or added as Supplementary materials.

We added sequences in Supplementary materials Tab C see line 140.

7: Please specify in “Materials and Methods” section the software used for the analyses in supplementary figure A.

Supplementary material figure A and B were performed by using itol.embl.de/tree [Ci-tation: Letunic and Bork (2021) Nucleic Acids Res doi: 10.1093/nar/gkab301].

Other comments:

Line 179-182: are you sure that your typical haplotypes are not present in other Italian donkey breeds? For this reason, I suggest the alignment with other sequences from Italian donkey from GenBank.

Accordingly, we removed the world private and as already done, see major points, we perform alignment with GenBank

Line 189: please change “3.2.3. Origin and phyletic relationships” in “3.2.3. Origin and phylogenetic relationships”

done

Line 190: please change “The phyletic relationships…” in “The phylogenetic relationships…..”

done

Line 255: “….. iii. the analysis does not distinguish among breeds”.

It has been removed

We agree with the REV in biological terms of free ranging animals, but here we are in the same conditions of dog pure breed were official center perform let say ‘inbreeding’ between a given number of subject ascribable to a given breed; furthermore Pantesco is completely isolated from any other breed consequently as reported by Bannasch DL, Bannasch MJ, Ryun JR, Famula TR, Pedersen NC (2005) Y chromosome haplotype analysis in purebred dogs. Mamm Genome 16(4):273-80. doi: 10.1007/s00335-004-2435-8: “Breed-specific haplotypes were identified for 26 of the 50 breeds, and haplotype sharing between some breeds indicated a common history, with a significant genetic variation across breeds and with geographic origin of the breeds, particularly among breeds of African origin.”

Sorry, but donkey bred is quite different from dog bred. Dog pure breeds are mainly selected using inbreeding matings with a high level of consanguinity and the breed differentiation is high, as reported in many papers using different markers (for example Sechi et al., 2016; https://doi.org/10.1080/1828051X.2016.1248867; Yang et al., 2019 https://doi.org/10.3389/fgene.2019.01174; Plassais et al., 2019 https://doi.org/10.1038/s41467-019-09373-w). The selective pressure operated by breeders is not so high in donkeys as in horses, so the haplotypes of mtDNA are shared and is impossible identify the breed by the haplotype, as well reported by Stanisic et al 2020 for the Banat donkey (Stanisic et al., 2020; https://doi.org/10.7717/peerj.8598).

In my opinion, the sentence have to be removed.

Line 283: please change “phyletic” with “phylogenetic”.

done

Line 286-391: Please clarify because the circular NJ in figure A is unclear, and I do not found evidence of a Chinese line and two African lines, Nubian and Somali, in your data.

It has been removed.

Line 317-318: This sentence is wrong and have to be removed. It is well known that, over time, PT and RG have repeatedly crossbred each other and with the MF or with the Catalan donkey.

 It has been removed.

Line 320-330: These sentences have to be move and match with “Conclusion” section or delete. Again, change “phyletic” with “phylogenetic”!

It was moved in the conclusion and changed.

We thank the reveiwer for the important contribution to the improvement of the article. All your concerns have been addressed.

1: In the previous review, I asked a table with polymorphic sites of the mtDNA sequences, according to the reference sequence X97337 and with absolute and relative haplotype frequencies. In your new version I do not found the table that is a mandatory point.

We apologize for this misunderstanding, according to the reviewer request, we added a table (new table 2) including absolute and relative haplotype frequencies.

2: Figure 2: the figure is really too small. I understand that the number refer to the SNPs identified, but they are not the real position on reference sequence X97337, so is impossible compare this figure with literature data. The figure have to be modify.

We thank the reviewer to highlight this point. We modified figure 2 in order to overcome these concerns: SNP numbers refer now to real position on the reference sequence. Plot and text were increased in size.

3: the Authors added sequence of African donkey Somalicus and Nubianus, Chinese ones and other Asiatic Equus kiang, E. hemionus and E. hemionus kulan, this is interesting but do not clarify the history of Italian donkeys. They should align the sequences of this study with those present in GenBank for the same breeds or for other Italian donkey breeds/populations, in order to improve the information about the genetic diversity of the Italian donkey breeds and in Catalonian donkey. I think this is mandatory and one of the goals of the paper.

We added the ClustalW alignment Supplementary materials with other Italian donkey, Tab B see line 133 and Fig B Italian alignment line 194

4: The reference sequence for the median-joining network analysis should be X97337, not the crossbred donkey. Please make again the analyses with the correct reference.

It was done using as reference X97337 which share the same haplotype with Ragusano donkey, now it has been clearly identify by x, see line 208. The reference sequence X97337 is indicated as x.

5: please add the GenBank accession numbers for your sequences.

We added in Supplementary materials the Submission ID # 2466755 Tab C see line 149

6: The sequences used for the NJ have to be listed in “Materials and Methods” section or added as Supplementary materials.

We added sequences in Supplementary materials Tab C see line 140.

7: Please specify in “Materials and Methods” section the software used for the analyses in supplementary figure A.

Supplementary material figure A and B were performed by using itol.embl.de/tree [Ci-tation: Letunic and Bork (2021) Nucleic Acids Res doi: 10.1093/nar/gkab301].

Other comments:

Line 179-182: are you sure that your typical haplotypes are not present in other Italian donkey breeds? For this reason, I suggest the alignment with other sequences from Italian donkey from GenBank.

Accordingly, we removed the world private and as already done, see major points, we perform alignment with GenBank

Line 189: please change “3.2.3. Origin and phyletic relationships” in “3.2.3. Origin and phylogenetic relationships”

done

Line 190: please change “The phyletic relationships…” in “The phylogenetic relationships…..”

done

Line 255: “….. iii. the analysis does not distinguish among breeds”.

It has been removed

We agree with the REV in biological terms of free ranging animals, but here we are in the same conditions of dog pure breed were official center perform let say ‘inbreeding’ between a given number of subject ascribable to a given breed; furthermore Pantesco is completely isolated from any other breed consequently as reported by Bannasch DL, Bannasch MJ, Ryun JR, Famula TR, Pedersen NC (2005) Y chromosome haplotype analysis in purebred dogs. Mamm Genome 16(4):273-80. doi: 10.1007/s00335-004-2435-8: “Breed-specific haplotypes were identified for 26 of the 50 breeds, and haplotype sharing between some breeds indicated a common history, with a significant genetic variation across breeds and with geographic origin of the breeds, particularly among breeds of African origin.”

Sorry, but donkey bred is quite different from dog bred. Dog pure breeds are mainly selected using inbreeding matings with a high level of consanguinity and the breed differentiation is high, as reported in many papers using different markers (for example Sechi et al., 2016; https://doi.org/10.1080/1828051X.2016.1248867; Yang et al., 2019 https://doi.org/10.3389/fgene.2019.01174; Plassais et al., 2019 https://doi.org/10.1038/s41467-019-09373-w). The selective pressure operated by breeders is not so high in donkeys as in horses, so the haplotypes of mtDNA are shared and is impossible identify the breed by the haplotype, as well reported by Stanisic et al 2020 for the Banat donkey (Stanisic et al., 2020; https://doi.org/10.7717/peerj.8598).

In my opinion, the sentence have to be removed.

Line 283: please change “phyletic” with “phylogenetic”.

done

Line 286-391: Please clarify because the circular NJ in figure A is unclear, and I do not found evidence of a Chinese line and two African lines, Nubian and Somali, in your data.

It has been removed.

Line 317-318: This sentence is wrong and have to be removed. It is well known that, over time, PT and RG have repeatedly crossbred each other and with the MF or with the Catalan donkey.

 It has been removed.

Line 320-330: These sentences have to be move and match with “Conclusion” section or delete. Again, change “phyletic” with “phylogenetic”!

It was moved in the conclusion and changed.